# WAVEN-PULL: WAVELET-BASED ANOMALY DETECTION IN DYNAMIC GRAPHS VIA POSITIVE-UNLABELED LEARNING

## ABSTRACT

Anomaly detection in dynamic graphs is vital for identifying evolving threats in domains such as social networks and financial systems. While Graph Neural Networks (GNNs) have shown promise, they typically require ample labeled data (both normal and anomalous) to distinguish anomalous nodes from normal ones. In practice, however, only a small subset of anomalies are labeled, leaving most nodes unlabeled, a scenario known as positive-unlabeled (PU) learning. This, combined with the oversmoothing tendency of GNNs, leads to a strong bias toward predicting most nodes as normal, thus severely limiting detection performance. To address these challenges, we propose WAVEN-PULL: WAVElet-based ANomaly detection in dynamic graphs via Positive-UnLabeled Learning. WAVEN-PULL features: (1) dynamic graph encoder that combines Beta-Wavelet Graph Convolution and temporal attention to capture multi-scale spectral patterns and the temporal evolution of node behaviors, thereby effectively capturing anomalous signals in dynamic graphs and mitigating oversmoothing; (2) PU-aware alignment module that corrects prediction bias by aligning the anomaly ratio of unlabeled predictions with class priors, which can be theoretically shown to yield an unbiased risk estimator with temporal stability under exponential moving average (EMA); and (3) anomaly probability estimation module that maps node embeddings to probabilities, ensuring consistency with risk minimization principles and enabling robust end-to-end detection even with scarce labels. Extensive experiments on real-world dynamic graph datasets demonstrate that WAVEN-PULL consistently outperforms state-of-the-art methods, achieving absolute AUC improvements of 6.15%, 22.81%, and 4.74% on Wikipedia, Reddit, and Bit-Alpha, respectively.

## 1 INTRODUCTION

Many real-world systems, such as social networks, financial transaction platforms, and traffic monitoring infrastructures, exhibit relational structures and node attributes that evolve over time, giving rise to what are known as dynamic graphs (Ekle & Eberle, 2024). In these scenarios, timely and accurate detection of anomalous nodes is crucial for ensuring system security, preventing the spread of risks, and optimizing resource allocation (Jiao et al., 2021). In contrast to static graphs, dynamic graphs exhibit continuously evolving spatiotemporal structures and node attributes, resulting in far more intricate and rapidly changing anomalous patterns. This ongoing evolution greatly increases the complexity of anomaly detection, making it a particularly challenging problem in dynamic graph settings (Guo et al., 2023; Liu et al., 2021).

Graph Neural Networks (GNNs) have shown strong performance in dynamic graph anomaly detection by learning effective node embeddings (Huang et al., 2022b;a; Wang et al., 2022). However, GNNs inherently suffer from over-smoothing: as layers or neighbor aggregations increase, node representations become overly similar (Duan et al., 2023; Tang et al., 2023), which is especially problematic for anomaly detection. The unique features of anomalous nodes are diluted by their normal neighbors, making anomalies hard to distinguish (Zhang et al., 2022; Zhou et al., 2023; Tang et al., 2023). This challenge is further exacerbated in the positive-unlabeled (PU) setting, where only a few anomalies are labeled and most nodes are unlabeled Ding et al. (2022); Thorpe et al. (2022). The abundance of unlabeled nodes biases models toward predicting all nodes as normal, leading to missed anomalies and model collapse(formally analyzed in Appendix A.3). Fundamentally, this

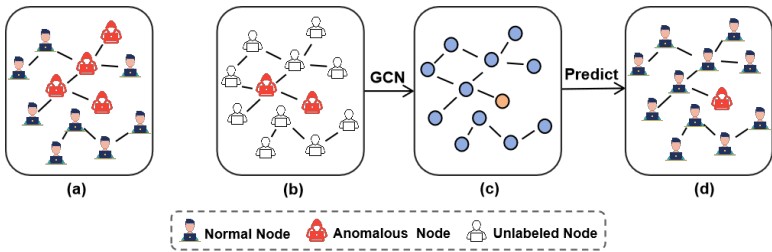

Figure 1: Illustration of the challenges in dynamic anomaly detection under sparse supervision, taking a single graph snapshot as an example. (a) shows the ground-truth anomaly states; (b) reflects the realistic PU setting with only a few known anomalies; (c) visualizes the over-smoothing effect of GCNs, where the anomalous node's embedding becomes indistinguishable; and (d) depicts the biased prediction outcome where most nodes are predicted as normal.

is because GNN aggregation acts as a low-pass filter(see Proposition A.3 for the formal proof): it preserves smooth, low-frequency patterns while suppressing the sharp, high-frequency signals that often indicate anomalies. In the PU setting, this suppression of high-frequency information is even more severe, as the model is further biased to ignore these crucial components (see our analysis of this coupled amplification in Appendix A.3.3). Therefore, a frequency-domain approach that explicitly preserves high-frequency information is needed to counteract over-smoothing and supervision bias. However, existing frequency-based anomaly detection methods are typically designed for static graphs, cannot handle temporal dynamics, and lack robustness to the supervision bias inherent in the PU setting, making them ineffective for real-world, sparsely labeled dynamic graphs.

To address these challenges, we propose: **WAVE**let-based A**N**omaly detection in dynamic graphs via **P**ositive-**U**n**L**abeled **L**earning (WAVEN-PULL), a unified framework that jointly preserves anomaly signals and ensures unbiased supervision. Our core insight is that effective anomaly detection in dynamic graphs requires both node representations that retain high-frequency and temporal information, and a learning objective that avoids bias from scarce labels. WAVEN-PULL integrates two tightly coupled modules: (1) a dynamic graph encoder that uses Beta-Wavelet Graph Convolution to preserve high-frequency structural features and temporal attention to capture evolving node behaviors, and (2) a PU-aware calibration module that aligns predictions on unlabeled data with the known anomaly prior, providing stable, unbiased supervision without requiring negative labels. This joint optimization prevents over-smoothing and model collapse, enabling robust detection of anomalies even under extreme label scarcity.

We highlight the key contributions of this work as follows:

- We formalize dynamic graph anomaly detection under the challenging yet practical Positive-Unlabeled (PU) setting, and provide a theoretical analysis revealing how the interplay between GNN over-smoothing and PU supervision bias fundamentally drives model collapse (see Appendix A.3.3). Our analysis establishes the necessity of frequency-aware and bias-corrected approaches in this setting.

- We introduce WAVEN-PULL, a unified framework with two theoretically grounded modules: (1) a dynamic graph encoder leveraging Beta-Wavelet Graph Convolution and temporal attention to provably preserve high-frequency and temporal anomaly signals (see Theorem B.5 for the formal proof); and (2) a PU-aware learning objective that aligns predictions on unlabeled data with the anomaly class prior, ensuring unbiased and stable training even without negative labels. We provide formal guarantees for the optimality, stability, and robustness of our objective (see Proposition 4.1 and related lemmas).

- Extensive experiments on real-world dynamic graph benchmarks demonstrate the effectiveness of WAVEN-PULL, which consistently surpasses state-of-the-art baselines. Notably, WAVEN-PULL achieves up to 6.15%, 32.2%, and 11.34% absolute improvements in AUC, Precision@50, and Recall@50, respectively, validating both our theoretical insights and practical design.

## 2   RELATED WORK

In the real world, the structure, attributes, and interactions of networks may evolve over time, forming dynamic graphs. Dynamic graph anomaly detection seeks to identify nodes, edges, or

subgraphs whose behaviors deviate significantly from normative patterns over time. Compared to static graphs, dynamic graphs pose greater challenges for anomaly detection: anomalies may only be prominent within specific time intervals, or they may become hidden or transform as time passes. This dynamic nature makes it difficult to directly transfer static graph anomaly detection methods, as they often fail to capture temporal signals and complex abnormal patterns induced by evolving topologies. A series of recent methods have focused on dynamic graph representation learning, providing a basis for anomaly detection in dynamic networks. For example, TGAT Xu et al. (2020) uses self-attention layers and temporal encoding to aggregate neighborhood features and learn dynamic representations, and TGN Rossi et al. (2020) integrates a memory module to capture long-term dependencies and employs embedding modules to mitigate memory staleness, combined with efficient parallel processing and attention-based graph aggregation. However, when applied to anomaly detection, these approaches often still rely on partially available labeled anomalies, and the high cost of obtaining such labels limits their practical applicability.

To tackle this challenge, SAD Tian et al. (2023) introduced an end-to-end semi-supervised anomaly detection framework that combines a time-equipped memory bank with a pseudo-label contrastive learning module, aiming to enhance effectiveness in dynamic graph anomaly detection scenarios. TADDY Liu et al. (2021) explores a single Transformer model to incorporate both spatial and temporal information into node representations of dynamic networks, but its heavy reliance on a single model structure may limit its generalization when facing diverse temporal distributions of anomalies. Moreover, there are works that seek to detect dynamic anomalies under limited or no labeled data conditions. NetWalk Yu et al. (2018) incrementally updates node embeddings and uses random walks to model dynamic graphs in an unsupervised manner, enabling the detection of dynamic anomalies without extensive labeling. AddGraph Zheng et al. (2019) employs an attention-based temporal GCN, fusing temporal and structural features for anomaly detection in dynamic graphs.

Furthermore, existing methods are often misaligned with the practical supervision constraints of real-world scenarios, which typically follow a Positive-Unlabeled (PU) setting. This challenge is distinct from standard learning paradigms. Semi-supervised frameworks, exemplified by SAD Tian et al. (2023), still struggle as they require a source of reliable negative or pseudo-labeled samples, a condition that is violated in a contaminated unlabeled set. Similarly, unsupervised approaches, such as NetWalk Yu et al. (2018), are founded on the assumption of a predominantly "normal" data distribution, a premise broken by both hidden anomalies and pervasive structural noise. To our knowledge, no existing work provides an integrated solution that is robust to structural noise under PU supervision, particularly by disentangling their overlapping effects in the spectral domain. This leaves a critical gap for a more realistic and robust detection framework, which our work aims to fill.

## 3 Preliminaries

In this section, we introduce the notation and formulate the problem statement. A quick background on key concepts is provided in Appendix C.

**Notation.** We use $\mathcal{G} = \{G_1, G_2, \ldots, G_T\}$ to denote a dynamic graph, where each snapshot $G_t = (V_t, E_t, \mathbf{X}_t)$, $t \in \{1, 2, \ldots, T\}$, represents the state of the graph at time step $t$. Here, $V_t = \{v_1^t, v_2^t, \ldots, v_n^t\}$ is the node set; $E_t \subseteq V_t \times V_t$ is the set of edges; and $\mathbf{X}_t \in \mathbb{R}^{|V_t| \times d}$ is the node feature matrix. Each node $v_i^t \in V_t$ may be associated with a latent label $y_i^t \in \{0, 1\}$, indicating whether it is anomalous (1) or normal (0). We summarize a list of commonly used symbols in Appendix (Table 7).

**Problem Statement.** We address anomaly detection in dynamic graphs under a Positive-Unlabeled (PU) setting, where only a few anomalous nodes are labeled and most nodes are unlabeled (potentially normal or anomalous). Given a dynamic graph $\mathcal{G} = \{G_1, \ldots, G_T\}$ with snapshots $G_t = (V_t, E_t, \mathbf{X}_t)$, we have a small set of labeled anomalies $V^L$ and treat all other nodes as unlabeled $V^U = \bigcup_{t=1}^{T} V^t \setminus V^L$. The objective is to learn a scoring function $f_\theta : V_t \to \mathbb{R}$ that assigns each node $v \in V_t$ an anomaly score $s_v^t$, indicating its likelihood of being anomalous at time $t$.

## 4 Waven-Pull

**Overview.** The overview of Waven-Pull is illustrated in Figure 2, which consists of three key modules: **(i) Dynamic graph encoding module** that captures low-frequency structural patterns and high-frequency anomaly signals through Beta-Wavelet graph convolution, and models temporal

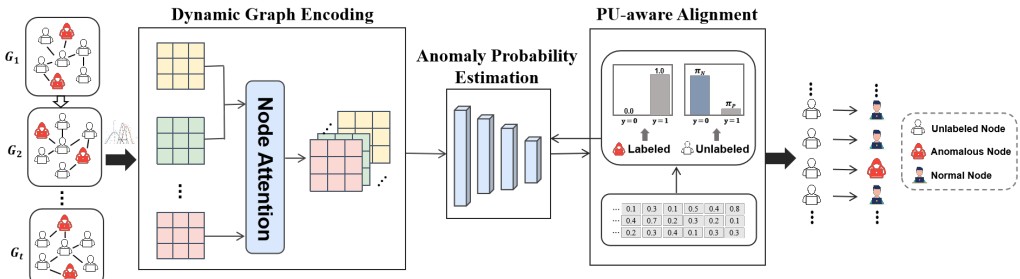

Figure 2: Overall workflow of WAVEN-PULL.

dynamics of node behavior via node-level temporal attention; **(ii) PU-aware alignment module** to address label scarcity and class imbalance by calibrating predicted anomaly distributions to match a prior and stabilizing training with temporal smoothing; **(iii) Anomaly probability estimation module** to map node representations to anomaly probabilities via a feedforward network with skip connections. All modules are trained end-to-end for robust anomaly detection under limited supervision.

### 4.1 DYNAMIC GRAPH ENCODING MODULE

We extend the Beta-Wavelet Graph Neural Network (BWGNN) Tang et al. (2022) with temporal modeling to encode dynamic graphs for anomaly detection. Our approach decomposes node features into spectral components, where high-frequency signals highlight abrupt structural deviations (potential anomalies), and low-frequency components capture the smooth background structure. This spectral decomposition is combined with temporal attention to model evolving node behaviors.

**Spectral Decomposition.** For each snapshot $G_t$ of the dynamic graph $\mathcal{G}$, we construct the adjacency matrix $A_t$ (with self-loops) and degree matrix $D_t = \text{diag}(A_t \mathbf{1})$. The symmetric normalized Laplacian is:

$$L_t = I - (D_t)^{-1/2} A_t (D_t)^{-1/2}. \tag{1}$$

This Laplacian underpins our spectral filtering. To extract multi-frequency features, we use a bank of Beta-wavelet polynomial filters (He et al., 2021) of order $C$:

$$g_{p,q}(\lambda) = \frac{1}{2B(p+1, q+1)} \left(\frac{\lambda}{2}\right)^p \left(1 - \frac{\lambda}{2}\right)^q, \tag{2}$$

for $p, q \geq 0$ with $p + q = C$, where $B(\cdot, \cdot)$ is the Beta function. Each $g_{p,q}(\lambda)$ acts as a band-pass filter centered at $\lambda_{\text{peak}} = 2p/C$, enabling selective emphasis of spectral regions.

*Theoretical connection.* As shown in Appendix A (Lemma A.1), the energy of a graph signal $x$ decomposes as $\sum_i \lambda_i \hat{x}_i^2$ across frequencies. Applying our spectral filter $g_{p,q}$ yields filtered coefficients $\widehat{(g_{p,q} * x)}_i = g_{p,q}(\lambda_i)\hat{x}_i$ (Appendix B.1), reweighting the energy by $g_{p,q}(\lambda_i)^2$. Unlike standard $K$-layer GNNs, which are proven to suppress high-frequency anomaly signals (Lemma A.2 and Proposition A.3), our filter bank covers the full spectrum. This preserves anomaly-relevant high-frequency energy, a property that is formally established in our main theoretical analysis (Proposition B.2 and Corollary B.3).

**Efficient Filtering.** Since $g_{p,q}(\lambda)$ is a degree-$C$ polynomial, we avoid eigendecomposition by applying the filter as a matrix polynomial:

$$\mathcal{W}_{p,q}(A_t) = g_{p,q}(L_t) = \frac{1}{2B(p+1, q+1)} \left(\frac{L_t}{2}\right)^p \left(I - \frac{L_t}{2}\right)^q. \tag{3}$$

Given node features $\mathbf{X}_t \in \mathbb{R}^{|V^{(t)}| \times F_{\text{in}}}$, we compute filtered features:

$$\mathbf{Z}_t^{(p)} = \mathcal{W}_{p,q}(A_t) \mathbf{X}_t \mathbf{W}^{(p)}, \qquad p = 0, \ldots, C, \tag{4}$$

where $\mathbf{W}^{(p)}$ are learnable projections. The multi-scale spatial embedding is:

$$\mathbf{H}_t^{\text{sp}} = [\mathbf{Z}_t^{(0)} \| \cdots \| \mathbf{Z}_t^{(C)}] \mathbf{W}_{\text{mix}} \in \mathbb{R}^{|V^{(t)}| \times d}. \tag{5}$$

**Temporal Attention.** To capture temporal dynamics, we apply node-level attention over previous snapshots $k < t$:

$$Q^t = \mathbf{H}_t^{\text{sp}}\mathbf{W}_Q, \quad K^k = \mathbf{H}_k^{\text{sp}}\mathbf{W}_K, \quad V^k = \mathbf{H}_k^{\text{sp}}\mathbf{W}_V, \tag{6}$$

$$\alpha_k = \texttt{softmax}(Q^t K^k / \sqrt{d}), \tag{7}$$

$$\mathbf{H}_t^{\text{att}} = \sum_{k=1}^{t-1} \alpha_k V^k, \tag{8}$$

where $W_Q, W_K, W_V$ are learnable. The final node representation is:

$$\mathbf{H}_t = \mathbf{H}_t^{\text{sp}} + \mathbf{H}_t^{\text{att}}. \tag{9}$$

*Theoretical motivation.* As established in Appendix A.2.2, performing spectral analysis on individual graph snapshots is fundamentally susceptible to temporal blind spots and instability arising from nonstationary dynamics (see Lemma A.4). To address these deficiencies, our temporal attention mechanism is explicitly constructed to incorporate historical information, thereby mitigating the limitations of snapshot-based approaches. Importantly, this temporal aggregation is rigorously shown to preserve the essential spectral characteristics produced by the spatial encoder, as it does not induce any additional spatial smoothing; this property is formally proven in Appendix B.1.4 (Lemma B.4). The overall efficacy of our joint encoder is substantiated by our principal theoretical result in Appendix B.1.5. In particular, Theorem B.5 demonstrates that the resulting representation $\mathbf{H}_t$ exhibits a strictly reduced over-smoothing index ($\text{OS}_t$), thereby providing a formal guarantee that our module effectively mitigates the over-smoothing phenomenon.

### 4.2 PU-AWARE ALIGNMENT MODULE

WAVEN-PULL addresses the challenges of label sparsity and class imbalance in dynamic PU anomaly detection by enforcing two key mechanisms: **(a) Label Distribution Alignment** to match the predicted anomaly rate to the class prior, and **(b) Temporal Stabilization via Exponential Moving Average (EMA)** to smooth predictions over time. These strategies mitigate majority-class bias and temporal noise, leading to more robust PU anomaly detection.

**(a) Label Distribution Alignment.** In standard binary classification, the expected risk for a classifier $f$ is

$$R(f) = \mathbb{E}_{(\boldsymbol{x},y)\sim p(\boldsymbol{x},y)}\left[\mathbb{1}\left[\hat{y} \neq y\right]\right], \tag{10}$$

where $\hat{y} = \mathbb{1}[f(x) \geq 0]$. This decomposes as

$$\begin{aligned} R(f) &= \pi_P \left|\mathbb{E}_{\boldsymbol{x}\sim p_P(\boldsymbol{x})}[\hat{y}] - 1\right| + \pi_N \left|\mathbb{E}_{\boldsymbol{x}\sim p_N(\boldsymbol{x})}[\hat{y}] - 0\right| \\ &= \pi_P R_P + \pi_N R_N, \end{aligned} \tag{11}$$

where $\pi_P$ and $\pi_N = 1 - \pi_P$ are class priors, and $R_P, R_N$ are the risks on positives and negatives, respectively.

In the PU setting, negatives are unlabeled, but the expected label for an unlabeled sample is $\pi_P$. The expected prediction over unlabeled data is

$$\mathbb{E}_{\boldsymbol{x}\sim p_U(\boldsymbol{x})}[\hat{y}] = \pi_P \mathbb{E}_{\boldsymbol{x}\sim p_P(\boldsymbol{x})}[\hat{y}] + \pi_N \mathbb{E}_{\boldsymbol{x}\sim p_N(\boldsymbol{x})}[\hat{y}]. \tag{12}$$

This allows us to express the negative risk $R_N$ in terms of observable quantities:

$$\pi_N R_N = \mathbb{E}_{\boldsymbol{x}\sim p_U(\boldsymbol{x})}[\hat{y}] - \pi_P + \pi_P R_P. \tag{13}$$

Substituting into equation 11 yields

$$R(f) = 2\pi_P R_P + \mathbb{E}_{\boldsymbol{x}\sim p_U(\boldsymbol{x})}[\hat{y}] - \pi_P. \tag{14}$$

However, both this objective and simpler heuristics are susceptible to the supervision bias inherent in the PU setting, where the optimization is dominated by the vast unlabeled set. This bias creates a persistent pressure to decrease predictions on all unlabeled nodes, leading to model collapse. Appendix A.3 provides a formal analysis of this failure mode using a canonical naive objective as a case study. To create an objective that is robust to this supervision bias, we introduce a label alignment penalty that directly enforces consistency with the known class prior :

$$R_{\text{lab}}(f) = 2\pi_P R_P + \left|\mathbb{E}_{\boldsymbol{x}\sim p_U(\boldsymbol{x})}[\hat{y}] - \pi_P\right|. \tag{15}$$

This alignment term $R_U$ removes the degenerate descent direction by anchoring the mean prediction on unlabeled data to $\pi_P$. This approach is mathematically justified in Appendix B.2, where we prove its equivalence to solving the theoretically optimal Neyman-Pearson classification problem (see Proposition B.9 and Theorem B.11).

In practice, we estimate the expectations empirically: for labeled anomalies $V^L$ and unlabeled nodes $V^U$,

$$\mathbb{E}[\hat{y}_P] = \frac{1}{|V^L|} \sum_{v \in V^L} \hat{y}_v, \quad \mathbb{E}[\hat{y}_U] = \frac{1}{|V^U|} \sum_{v \in V^U} \hat{y}_v.$$

Thus, the risk becomes

$$R_{\text{lab}}(f) = 2\pi_P \left| \mathbb{E}[\hat{y}_P] - 1 \right| + \left| \mathbb{E}[\hat{y}_U] - \pi_P \right|.$$

This formulation directly counteracts the bias and collapse issues of the naive PU objective (see Appendix B.2 for detailed analysis).

**(b) Temporal Stabilization via EMA.** Dynamic graphs introduce temporal noise in anomaly predictions, especially for the mean prediction over unlabeled nodes. To stabilize training, we apply an exponential moving average (EMA) to the estimated anomaly rate:

$$\hat{\mu}_t = \alpha_U \hat{\mu}_{t-1} + (1 - \alpha_U)\mathbb{E}[\hat{y}_U], \tag{16}$$

where $\alpha_U \in [0, 1]$ controls smoothing. At $t = 1$, we set $\hat{\mu}_1 = \mathbb{E}[\hat{y}_U^{(1)}]$. We then use $\hat{\mu}_t$ in place of the raw batch estimate in the risk:

$$R_{\text{final}}(f) = 2\pi_P \left| \mathbb{E}[\hat{y}_P] - 1 \right| + \left| \hat{\mu}_t - \pi_P \right|. \tag{17}$$

Mathematically, EMA reduces the effect of transient errors: letting $e_t = \mathbb{E}[\hat{y}_U] - \pi_P$ and $\tilde{e}_t = \hat{\mu}_t - \pi_P$, we have

$$\tilde{e}_t = \alpha_U \tilde{e}_{t-1} + (1 - \alpha_U)e_t,$$

so $|\tilde{e}_t| \leq \max\{|\tilde{e}_{t-1}|, |e_t|\}$, ensuring the alignment penalty is bounded by the worst of current and previous errors. This smoothing effect is crucial for stable optimization. As formally proven in Appendix B.2, the EMA operator provides three essential guarantees: stability against error amplification (Lemma B.13), long-term accuracy (Lemma B.14), and a quantifiable reduction in variance (Proposition B.15).

Finally, since the true class prior $\pi_P$ is typically unknown, we set it to a small, fixed value (e.g., $\pi_P = 0.01$) in our experiments. This is a common practice in anomaly detection scenarios where anomalies are inherently rare. This practical approach is principled, as our theoretical analysis in Appendix B.2 (Proposition B.16) formally proves that our framework is robust to misspecifications of the class prior. This robustness is further confirmed by our parameter sensitivity analysis in Appendix E.1.

### 4.3 Anomaly Probability Estimation Module

Each node representation $\mathbf{h}_i^{(t)}$ is mapped to an anomaly probability via a multi-layer feedforward network $f_\theta$ with skip connections, batch normalization, and dropout:

$$\hat{y}_i^{(t)} = \sigma(f_\theta(\mathbf{h}_i^{(t)})), \tag{18}$$

where $\sigma(\cdot)$ is the sigmoid function and $\hat{y}_i^{(t)} \in [0, 1]$. The network is trained end-to-end using the PU risk-based loss in equation 17, ensuring that anomaly scores are both label- and temporally-aligned as justified above and in the Appendix B.2.

Our design is not only empirically effective but also theoretically grounded. Specifically, our comprehensive analysis in Appendix B establishes rigorous guarantees for the core components of WAVEN-PULL, demonstrating the encoder's ability to preserve high-frequency signals and the learning objective's optimality and stability under PU supervision. We consolidate these key insights into the following proposition:

**Proposition 4.1** (Theoretical Guarantees of WAVEN-PULL). *Jointly optimizing the dynamic graph encoder and PU-aware alignment objective,* WAVEN-PULL *provably (i) preserves high-frequency structural signals, (ii) attains an optimal and stable PU classifier under prior alignment, and (iii) achieves robustness against oversmoothing and model collapse in dynamic graphs.*

Formal derivations and proofs are deferred to Appendix B.3.3.

### 4.4 WAVEN-PULL INTEGRATION

We unify the above components into an end-to-end framework. We proceed as follows: (1) We first extract node embeddings $\{\mathbf{H}_1, \ldots, \mathbf{H}_T\}$ from dynamic snapshots using Beta-Wavelet Graph Convolution and temporal attention. (2) For each training batch, we sample a few labeled anomalies from $V^L$ and more unlabeled nodes from $V^U$, forming $(\mathbf{X}_P, \mathbf{X}_U)$. (3) The model $f_\theta$ outputs anomaly scores, which are passed through a sigmoid to yield predictions $\hat{y}_v$. (4) The loss combines: (a) a labeled term $R_P = |\mathbb{E}[\hat{y}_P] - 1|$ to encourage high scores for labeled anomalies, and (b) an unlabeled term $R_U = |\alpha_U \hat{\mu}_{t-1} + (1 - \alpha_U)\mathbb{E}[\hat{y}_U] - \pi_P|$ to align the predicted anomaly rate among unlabeled nodes with the class prior $\pi_P$ using EMA smoothing. The total loss is:

$$\mathcal{L}_{\text{total}} = 2\pi_P R_P + R_U. \tag{19}$$

We train the model by minimizing $\mathcal{L}_{\text{total}}$ over all epochs. This approach ensures robust and temporally stable anomaly detection under PU supervision. The full training process is detailed in Algorithm 1.

## 5 EXPERIMENTATION

### 5.1 EXPERIMENTAL SETTINGS

**Datasets.** We employ four real-world datasets, **Wikipedia** Kumar et al. (2019), **Reddit** Kumar et al. (2019), **Bit-Alpha** Lee et al. (2024) and **Bit-OTC** Lee et al. (2024). The statistics are shown in Table 4, and their detailed introduction can be found in Appendix D.1. For all datasets, we divide the dataset into five temporal snapshots and reserve 90% of the final snapshot for testing. The remaining data is split 80:20 for training and validation. To simulate the Positive–Unlabeled (PU) learning setting, we randomly sample 100 anomalies from the training set as labeled positives. All remaining samples—including both normal nodes and unlabeled anomalies—are treated as unlabeled data during training.

**Baseline Methods.** To evaluate the performance, we include several state-of-the-art methods as baselines. These include: TGAT Xu et al. (2020), TGN Rossi et al. (2020), GDN Ding et al. (2021), AMNet Chai et al. (2022), SAD Tian et al. (2023), and SLADE Lee et al. (2024). Further details are explained in Appendix D.2.

**Evaluation Metric.** We evaluate model performance using three widely adopted metrics: AUCDing et al. (2021); Wang et al. (2021), Precision@KYuan et al. (2023), and Recall@KYuan et al. (2023). Further details and mathematical formulations of the metrics are provided in Appendix D.3.

**Experimental Setup.** Detailed hyperparameter configurations and implementation details are provided in Appendix D.4.

### 5.2 MAIN RESULTS

We report the overall performance of WAVEN-PULL in Table 1 (AUC) and Table 2 (Precision@K and Recall@K), comparing it against several state-of-the-art baselines across four dynamic graph benchmarks. In all tables, we boldface the best performing scores and underline the second-best results. For the AUC metric, WAVEN-PULL achieves the best performance on Wikipedia, Reddit, and Bit-Alpha, and ranks second on Bit-OTC with an AUC of 0.7353, closely following the unsupervised baseline SLADE (0.7718). The relative AUC improvements over the strongest baseline reach 6.1% on Wikipedia, 22.8% on Reddit, and 4.7% on Bit-Alpha. These gains stem from two key design elements: (1) our dynamic encoder, which combines Beta-Wavelet Graph Convolution and node-level temporal attention to capture both high-frequency anomaly cues and temporal node dynamics; and (2) our PU-aware label distribution alignment module, which corrects prediction bias under extreme label sparsity. Compared to existing methods, TGAT and TGN do not explicitly address anomaly detection or label imbalance, leading to lower performance. AMNet incorporates spectral filters but lacks temporal modeling, while SLADE performs well on Bit-OTC (a dataset with bursty edges) yet underperforms on others, highlighting the benefits of leveraging even a small number of positive labels. In Table 2, WAVEN-PULL also consistently outperforms baselines on ranking-based metrics. For instance, on Precision@100, it improves over the strongest baseline by 34.7% on Wikipedia, 61.1% on Reddit, 4.0% on Bit-Alpha, and 0.7% on Bit-OTC. Similarly, substantial gains in Recall@100 are observed, especially on Reddit and Wikipedia.The unsupervised baseline SLADE achieves strong AUC scores but struggles to prioritize true anomalies, as indicated by its lower Precision@K and Recall@K. In contrast, AMNet, despite its lower AUC, performs competitively on the Bitcoin datasets, with some @K metrics ranking among the best. These observations highlight the value of frequency-domain signals in anomaly detection and support the design rationale behind our method, which integrates frequency- and time-aware modeling under PU supervision.

Table 1: Performance Comparison of Different Methods Across Datasets. We report the AUC scores for WAVEN-PULL compared with other baselines.

| Method | Wikipedia | Reddit | Bit-Alpha | Bit-OTC |
|---|---|---|---|---|
| TGAT | 0.8563 | 0.6664 | 0.6227 | 0.6615 |
| TGN | 0.8753 | 0.6673 | 0.6602 | 0.5254 |
| GDN | 0.8512 | 0.6782 | 0.7687 | 0.6227 |
| AMNet | 0.5423 | 0.4966 | 0.6447 | 0.6960 |
| SAD | 0.8554 | 0.6789 | 0.7489 | 0.6498 |
| SLADE | 0.8868 | 0.7508 | 0.7692 | **0.7718** |
| WAVEN-PULL | **0.9483** | **0.9789** | **0.8166** | 0.7353 |

Table 2: Precision@50(P@50) and Recall@50(R@50) comparison with baselines across four datasets.

| Method | Wikipedia | | Reddit | | Bit-Alpha | | Bit-OTC | |
|---|---|---|---|---|---|---|---|---|
| | **P@50** | **R@50** | **P@50** | **R@50** | **P@50** | **R@50** | **P@50** | **R@50** |
| TGAT | 0.0042 | 0.0227 | 0.0040 | 0.0426 | 0.2333 | 0.0365 | 0.2453 | 0.0277 |
| TGN | 0.0000 | 0.0000 | 0.0280 | 0.0149 | 0.0760 | 0.0165 | 0.0560 | 0.0060 |
| GDN | 0.0280 | 0.0318 | 0.0000 | 0.0000 | 0.1400 | 0.0304 | 0.2500 | 0.0266 |
| AMNet | 0.0220 | 0.0289 | 0.0400 | 0.0250 | 0.1220 | 0.1355 | 0.3580 | 0.1125 |
| SAD | 0.0140 | 0.0159 | 0.0060 | 0.0032 | 0.1400 | 0.0304 | 0.2780 | 0.0296 |
| SLADE | 0.0200 | 0.0227 | 0.0000 | 0.0000 | 0.3600 | 0.0783 | 0.3800 | 0.0404 |
| WAVEN-PULL | **0.3600** | **0.1452** | **0.7600** | **0.2550** | **0.3800** | **0.7600** | **0.4400** | **0.2136** |

Overall, these results validate the importance of (i) frequency–temporal joint modeling and (ii) PU-aware distribution alignment, both of which are central to the design of WAVEN-PULL and are key drivers of its superior anomaly detection performance across diverse dynamic graph benchmarks.

We conduct a comprehensive set of experiments to evaluate WAVEN-PULL from multiple perspectives. Below, we present the ablation results and sample setting analysis in the main text. Additional experimental analyses, including robustness analysis, error analysis and few shot evaluation, are provided in Appendix E.

### 5.3 ABLATION STUDY

For ablation analysis, we systematically disable or replace one key module at a time to create four distinct model variants. The results, shown in Figure 3, compare the AUC, Precision@50, and Recall@50 metrics for each variant, highlighting the contribution of each component to the overall performance. We summarize our core findings as follows:

**w/o Temporal Attention (–TA).** This variant removes the temporal attention mechanism and replaces it with mean pooling over historical embeddings. The performance drop demonstrates the importance of modeling temporal evolution in dynamic graphs. Temporal attention enables the model to adaptively weigh past snapshots based on their relevance, allowing it to capture key changes in node behavior over time. In contrast, mean pooling treats all historical states equally, overlooking critical temporal shifts and weakening anomaly detection.

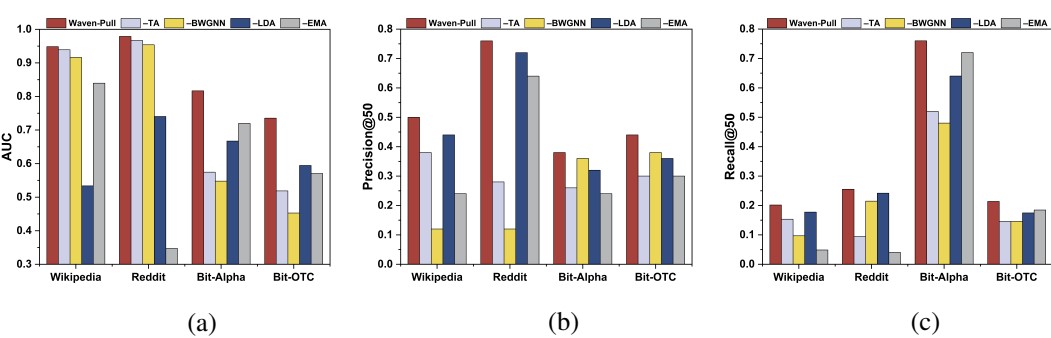

(a)         (b)         (c)

Figure 3: Performance comparison of WAVEN-PULL and its variants after removing key components, evaluated by (a) AUC, (b) Precision@50, and (c) Recall@50.

Table 3: AUC, Precision@50, and Recall@50 under varying numbers of labeled anomalies.

| | Wikipedia | | | Reddit | | | Bit-Alpha | | | Bit-OTC | | |
|---|---|---|---|---|---|---|---|---|---|---|---|---|
| Num | AUC | P@50 | R@50 | AUC | P@50 | R@50 | AUC | P@50 | R@50 | AUC | P@50 | R@50 |
| 50 | 0.9431 | 0.3600 | 0.1452 | 0.9782 | 0.7200 | 0.2416 | 0.7381 | 0.3400 | 0.6800 | 0.7449 | 0.4600 | 0.2233 |
| 60 | 0.9383 | 0.3000 | 0.1210 | 0.9788 | 0.8000 | 0.2685 | 0.7815 | 0.3800 | 0.7600 | 0.7353 | 0.3800 | 0.1845 |
| 70 | 0.9436 | 0.4000 | 0.1613 | 0.9785 | 0.7400 | 0.2483 | 0.7687 | 0.3200 | 0.6400 | 0.7616 | 0.5400 | 0.2621 |
| 80 | 0.9432 | 0.3400 | 0.1371 | 0.9788 | 0.7400 | 0.2483 | 0.7959 | 0.3600 | 0.7200 | 0.7573 | 0.5000 | 0.2427 |
| 90 | 0.9450 | 0.4000 | 0.1613 | 0.9793 | 0.7800 | 0.2617 | 0.8140 | 0.3800 | 0.7600 | 0.7546 | 0.4600 | 0.2233 |
| 100 | 0.9483 | 0.5000 | 0.2016 | 0.9789 | 0.7600 | 0.2550 | 0.8166 | 0.3800 | 0.7600 | 0.7353 | 0.4400 | 0.2136 |
| 110 | 0.9464 | 0.3800 | 0.1532 | 0.9792 | 0.7400 | 0.2483 | 0.8170 | 0.3600 | 0.7200 | 0.7477 | 0.4600 | 0.2233 |
| 120 | 0.9423 | 0.4200 | 0.1694 | 0.9792 | 0.7600 | 0.2550 | 0.8200 | 0.3800 | 0.7600 | 0.7440 | 0.5000 | 0.2427 |
| 130 | 0.9459 | 0.4200 | 0.1694 | 0.9802 | 0.8000 | 0.2685 | 0.8245 | 0.3600 | 0.7200 | 0.7402 | 0.4400 | 0.2136 |
| 140 | 0.9414 | 0.2600 | 0.1048 | 0.9796 | 0.8400 | 0.2819 | 0.8287 | 0.4000 | 0.8000 | 0.7482 | 0.5000 | 0.2427 |
| 150 | 0.9465 | 0.4400 | 0.1774 | 0.9798 | 0.8200 | 0.2752 | 0.8362 | 0.4000 | 0.8000 | 0.7492 | 0.5000 | 0.2427 |

**w/o Beta-Wavelet Graph Convolution (–BWGNN).** This variant substitutes the Beta-Wavelet Graph Convolution with a standard GCN layer. The resulting decline in AUC and Precision@50 underscores the value of frequency-aware modeling. The wavelet design captures high-frequency components associated with localized, subtle anomalies. In contrast, standard GCNs act as low-pass filters, oversmoothing node features and suppressing anomaly signals.

**w/o Label Distribution Alignment (–LDA).** In this ablation study, we replace our label distribution alignment loss with the standard binary cross-entropy (BCE) loss, where all unlabeled nodes are directly treated as normal. The resulting drops in AUC, Precision@50, and Recall@50 indicate impaired anomaly detection, as the weak signal from the few true anomalies is overwhelmed by the noisy, majority-class signal from the unlabeled set. This increases overfitting to the majority class and weakens detection of rare anomalies. These findings underscore the essential role of label alignment in providing robust supervision when confirmed positive labels are extremely scarce.

**w/o Exponential Moving Average (–EMA).** In this variant, we remove the EMA-based smoothing and instead update the anomaly expectation solely based on each mini-batch. This leads to increased volatility in both AUC and Recall@50 metrics, highlighting the crucial role of temporal smoothing. The EMA mechanism helps to dampen short-term prediction noise and stabilize the learning process, which is especially important in the presence of noisy labels and limited supervision. Without it, the model is more vulnerable to transient fluctuations, leading to less consistent estimation of anomaly distributions over time.

### 5.4 SAMPLE SETTING ANALYSIS

In our setting, only a small subset of anomalies are labeled, with the vast majority of nodes remaining unlabeled during training, This scenario reflects the severe label sparsity typical in real-world anomaly detection. To assess how well WAVEN-PULL copes with varying levels of supervision, we vary the number of labeled anomalies (*num_labeled*) from 50 to 150. The **Num** column in Table 3 indicates the value of *num_labeled* for each experiment. As shown in Table 3, WAVEN-PULL demonstrates robust and stable performance across all datasets, even when supervision is extremely limited. With as few as 50 labeled anomalies, the model achieves AUC scores above 0.94 on Wikipedia and Reddit, and above 0.74 on the Bitcoin datasets. As the number of labeled anomalies increases, we observe gradual improvements, especially in top-ranked precision and recall, but the overall performance remains strong even in the lowest-label regime. This resilience is attributed to our joint frequency–temporal modeling and label distribution alignment, which together help mitigate prediction bias and enhance the detection of subtle anomalies under scarce supervision—a crucial property for practical deployment where large-scale labeling is rarely feasible.

## 6 CONCLUSION

This work addresses the critical challenge of prediction bias in dynamic graph anomaly detection, a problem rooted in the realistic yet underexplored positive-unlabeled (PU) setting where only a few anomalies are labeled. Our method WAVEN-PULL combines a frequency-aware spatial encoder with temporal attention to capture evolving high-frequency anomaly signals, and a PU-aware alignment module provides stable supervision, preventing the model from collapsing to majority-class predictions. Extensive experiments on real-world dynamic graph datasets demonstrate that WAVEN-PULL consistently outperforms state-of-the-art methods under limited supervision scenarios, highlighting its robustness and potential applicability in various realistic anomaly detection tasks.

ETHICS STATEMENT

We affirm adherence to the ICLR Code of Ethics. Our work studies algorithmic methods for dynamic graph anomaly detection under positive–unlabeled setting, intended for beneficial applications such as fraud/abuse detection, infrastructure monitoring, and network security. We acknowledge potential risks of misuse (e.g., surveillance, unfair treatment, or privacy violations). Our experiments rely exclusively on publicly available datasets used under their licenses; we collect no new data, handle no personally identifiable information, and involve no human subjects (IRB not applicable). Any deployment should include human oversight, task-specific thresholding, careful prior estimation, and periodic bias/drift audits, and must comply with applicable privacy and data-protection laws. We disclose no conflicts of interest or sensitive sponsorship.

REPRODUCIBILITY STATEMENT

We are committed to ensuring the reproducibility of our research. All datasets used in our experiments are publicly available, and we provide detailed descriptions of data preprocessing, experimental settings, and hyperparameters in Appendix D. To facilitate full verification of our theoretical claims, complete proofs for all propositions and lemmas are included in Appendix A and Appendix B . The source code for our model, WAVEN-PULL, is not publicly available at the time of submission but will be made available upon reasonable request to the authors for purposes of academic verification and replication.

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

# A FUNDAMENTAL CHALLENGES IN DYNAMIC GRAPH ANOMALY DETECTION

## A.1 GNN OVER-SMOOTHING AS A LOW-PASS FILTER

### A.1.1 GNNS AS SPECTRAL FILTERS: A GRAPH SIGNAL PROCESSING PERSPECTIVE

Graph Neural Networks (GNNs) can be analyzed via graph signal processing by treating node features as signals on a graph, which enables a frequency-domain interpretation of aggregation.

Consider an undirected, nonnegative-weight graph $\mathcal{G} = (V, A, X)$ with $N$ nodes, adjacency $A \in \mathbb{R}^{N \times N}$, and node features $X \in \mathbb{R}^{N \times d}$. For simplicity, let $\mathbf{x} \in \mathbb{R}^N$ be a one-dimensional signal (the analysis applies column-wise to $X$). The (symmetric) normalized Laplacian is

$$L = I - D^{-1/2} A D^{-1/2},$$

with eigendecomposition

$$L = U \Lambda U^\top, \quad \Lambda = \mathrm{diag}(\lambda_1, \ldots, \lambda_N), \ \ 0 = \lambda_1 \le \cdots \le \lambda_N \le 2.$$

The Graph Fourier Transform (GFT) is $\hat{\mathbf{x}} = U^\top \mathbf{x}$. A linear GNN layer acts as a spectral filter $g(\lambda)$:

$$\mathbf{x}' = U g(\Lambda) U^\top \mathbf{x}, \qquad g(\Lambda) = \mathrm{diag}\big(g(\lambda_1), \ldots, g(\lambda_N)\big).$$

### A.1.2 LOW-PASS FILTERING AND OVER-SMOOTHING IN GNNS

**Smoothness functional.** We quantify the smoothness of $\mathbf{x}$ by the (normalized) Dirichlet energy $\mathbf{x}^\top L \mathbf{x}$. It penalizes edge-wise disagreements and admits two equivalent forms:

**Lemma A.1** (Dirichlet energy identities). *Let $L = I - D^{-1/2} A D^{-1/2}$ and set $y = D^{-1/2} \mathbf{x}$. Then*

$$\mathbf{x}^\top L \mathbf{x} = \tfrac{1}{2} \sum_{i,j} A_{ij} (y_i - y_j)^2 = \sum_{i=1}^N \lambda_i \, \hat{x}_i^2, \tag{20}$$

*where $L = U \Lambda U^\top$, $\hat{\mathbf{x}} = U^\top \mathbf{x}$.*

*Proof sketch.* Using $L = I - D^{-1/2} A D^{-1/2}$ and $y = D^{-1/2} \mathbf{x}$, $\mathbf{x}^\top L \mathbf{x} = \mathbf{x}^\top D^{-1/2}(D - A) D^{-1/2} \mathbf{x} = y^\top (D - A) y = \tfrac{1}{2} \sum_{i,j} A_{ij}(y_i - y_j)^2$. With $L = U \Lambda U^\top$ and $\hat{\mathbf{x}} = U^\top \mathbf{x}$, $\mathbf{x}^\top L \mathbf{x} = \hat{\mathbf{x}}^\top \Lambda \hat{\mathbf{x}} = \sum_i \lambda_i \hat{x}_i^2$. $\square$

Eq. equation 20 shows that (i) $\mathbf{x}^\top L \mathbf{x}$ is small when degree-normalized neighbors agree (edge-wise smoothness); and (ii) it weights spectral components by their eigenvalues, thus penalizing higher-frequency modes more heavily. Therefore, we quantify smoothness by

$$\mathbf{x}^\top L \mathbf{x} = \sum_{i=1}^N \lambda_i \hat{x}_i^2,$$

which connects directly to the low-pass behavior below.

**Lemma A.2** (Spectral Filter of a GCN Layer). *Ignoring non-linearities, a standard GCN layer using the renormalization trick ($\hat{S} = \tilde{D}^{-1/2} \tilde{A} \tilde{D}^{-1/2}$ with $\tilde{A} = A + I$) acts as a linear spectral filter on the graph signal $\mathbf{x}$. With respect to the associated Laplacian $\tilde{L} = I - \hat{S}$, this filter has a response function given by:*

$$g_{GCN}(\lambda) = 1 - \lambda.$$

*Proof.* The propagation rule of a linearized GCN layer is $\mathbf{x}' = \hat{S} \mathbf{x}$. By definition, $\hat{S} = I - \tilde{L}$. In the spectral domain defined by the eigendecomposition of $\tilde{L} = U \Lambda U^\top$, this operation becomes:

$$\mathbf{x}' = (I - \tilde{L})\mathbf{x} = U(I - \Lambda)U^\top \mathbf{x}.$$

This corresponds to a spectral filter with a diagonal operator $g(\Lambda) = I - \Lambda$, whose response function is $g(\lambda) = 1 - \lambda$. $\square$

The filter response $g_{\text{GCN}}(\lambda) = 1 - \lambda$ preserves low frequencies ($\lambda \approx 0$), but its effect becomes catastrophic when layers are stacked. The following proposition formalizes this phenomenon, which is the spectral essence of over-smoothing.

**Proposition A.3** (Over-smoothing as Multi-Layer Low-Pass Filtering)**.** *Stacking $K$ linearized GCN layers leads to over-smoothing, which manifests spectrally in two ways:*

> 1. ***Dirichlet Energy Reduction:*** *A single GCN layer smooths the signal by reducing its Dirichlet energy:* $(\mathbf{x}')^{\top} \tilde{L} \mathbf{x}' \le \mathbf{x}^{\top} \tilde{L} \mathbf{x}$.

> 2. ***Exponential Attenuation:*** *The $K$-layer composite filter has a response function $g_K(\lambda) = (1 - \lambda)^K$. This causes any non-zero frequency component ($\lambda \in (0, 2]$) to decay to zero as the number of layers $K \to \infty$.*

*Proof.* **Part 1 (Energy Reduction):** Let $\hat{\mathbf{x}}$ be the GFT of $\mathbf{x}$. The GFT of the output $\mathbf{x}'$ is $\hat{\mathbf{x}}' = (I - \Lambda)\hat{\mathbf{x}}$. The Dirichlet energy of $\mathbf{x}'$ is:

$$(\mathbf{x}')^{\top} \tilde{L} \mathbf{x}' = (\hat{\mathbf{x}}')^{\top} \Lambda \hat{\mathbf{x}}' = \sum_{i=1}^{N} \lambda_i (1 - \lambda_i)^2 \hat{x}_i^2.$$

Since $(1 - \lambda_i)^2 \le 1$ for all $\lambda_i \in [0, 2]$, this sum is less than or equal to $\sum_{i=1}^{N} \lambda_i \hat{x}_i^2 = \mathbf{x}^{\top} \tilde{L} \mathbf{x}$.

**Part 2 (Exponential Attenuation):** Stacking $K$ layers is equivalent to applying the propagation matrix $K$ times: $\mathbf{x}^{(K)} = (\hat{S})^K \mathbf{x}$. In the spectral domain, this corresponds to a filter with response $g_K(\lambda) = (g_{\text{GCN}}(\lambda))^K = (1 - \lambda)^K$. For any $\lambda \in (0, 2]$, $|1 - \lambda| < 1$, therefore $\lim_{K \to \infty}(1 - \lambda)^K = 0$. Only the zero-frequency component ($\lambda = 0$) is preserved. $\square$

### A.1.3 Implications for Anomaly Detection: A Fundamental Conflict

This low-pass property fundamentally conflicts with anomaly detection. Anomalous nodes have features $\mathbf{x}_i$ that differ from their neighbors, resulting in large gradients across edges and high values of $\mathbf{x}^T L \mathbf{x}$, i.e., energy in high-frequency components.

Mathematically, anomalies are encoded in the high-frequency spectrum:

$$\mathbf{x}^T L \mathbf{x} = \sum_{i=1}^{N} \lambda_i \hat{x}_i^2, \quad \text{with anomalies} \implies \hat{x}_i^2 \text{ large for } \lambda_i \gg 0.$$

However, GNNs, by design, suppress these high-frequency components, leading to "representational dilution": with each layer, anomaly-indicative signals are exponentially attenuated, and anomalous node embeddings become indistinguishable from normal ones. This analysis reveals a fundamental and inherent conflict: The objective of anomaly detection is to amplify high-frequency signals, yet the core architectural bias of message-passing GNNs is designed to suppress them. This explains the sub-optimal performance of standard GNNs for anomaly detection and motivates the need for architectures (e.g., graph wavelets) that can adaptively capture both low- and high-frequency information.

## A.2 The Failure of Static Filters on Dynamic Graphs

### A.2.1 Why a Frequency-Domain Approach is Feasible

Message-passing GNNs act as low-pass operators; stacking layers progressively suppresses high-frequency components and thus "washes out" anomaly-indicative signals. A more goal-aligned route is to work directly in the frequency domain and selectively amplify the high-frequency content that anomalies tend to carry.

This is not ad hoc. Both theoretical analyses and empirical observations support a *spectral right-shift*: as anomaly severity increases, energy moves from low to high graph frequencies. A convenient high-frequency summary is the Rayleigh quotient

$$S_{\text{high}}(\mathbf{x}, L) = \frac{\mathbf{x}^{\top} L \mathbf{x}}{\mathbf{x}^{\top} \mathbf{x}},$$

which grows as more energy concentrates in high-frequency modes and, crucially, can be computed without eigendecomposition—making it suitable for large graphs. Hence, continuing to use low-pass filters systematically attenuates precisely the information we wish to amplify; band-pass filtering is the more purposeful choice.

Graph wavelets provide an operational framework for this choice. Admissible wavelet kernels exhibit band-pass behavior in the spectral domain; a multi-scale bank of such kernels covers distinct frequency ranges; and, in practice, one implements these filters as polynomials of the Laplacian,

$$g(L) = \sum_{k=0}^{K} a_k L^k,$$

to avoid eigendecomposition and enable efficient, parallel sparse-matrix recurrences.

Compared with heat kernels (fundamentally low-pass), a well-designed wavelet bank offers both a low-pass channel and multiple band-pass channels. Multi-scale responses can be positive or negative, capturing not only neighborhood similarity but also explicit differences, which improves anomaly separability.

Moreover, appropriate wavelet families enjoy dual locality: (i) spectral locality, allowing the response to be anchored around target bands (so valuable high frequencies are not inadvertently smoothed away), and (ii) spatial locality, since a $K$-degree polynomial acts within $K$ hops, limiting global diffusion and mitigating the re-averaging that drives over-smoothing.

**Clarifying when snapshot scores become zero.** We make explicit the cases under which the snapshot high-frequency score $S_{\text{high}}(\mathbf{x}, L) = \frac{\mathbf{x}^\top L \mathbf{x}}{\mathbf{x}^\top \mathbf{x}}$ (and the band-pass captured energy) evaluate to zero.

**Lemma A.4** (Spatially constant signals yield zero snapshot score)**.**

1. **Unnormalized Laplacian** ($L = D - A$)**.** *If* $\mathbf{x}$ *is constant on each connected component (i.e.,* $\mathbf{x} = c\,\mathbf{1}$ *component-wise), then* $\mathbf{x}^\top L\,\mathbf{x} = 0$*, hence* $S_{\text{high}}(\mathbf{x}, L) = 0$*.*

2. **Symmetric normalized Laplacian** ($L_{\text{sym}} = I - D^{-1/2} A D^{-1/2}$)**.** *Let* $y = D^{-1/2}\mathbf{x}$*. If* $y$ *is constant on each connected component (equivalently,* $\mathbf{x} = D^{1/2} c\,\mathbf{1}$ *component-wise), then*

$$\mathbf{x}^\top L_{\text{sym}} \mathbf{x} = \tfrac{1}{2} \sum_{(i,j) \in \mathcal{E}} A_{ij} \left( y_i - y_j \right)^2 = 0,$$

*and thus* $S_{\text{high}}(\mathbf{x}, L_{\text{sym}}) = 0$*.*

*Proof sketch.* For (1), $L\mathbf{1} = \mathbf{0}$ on each connected component, so $\mathbf{x}^\top L\mathbf{x} = 0$ when $\mathbf{x}$ is constant. For (2), the edge-sum identity above shows the quadratic form is a sum of squared neighbor differences of $y = D^{-1/2}\mathbf{x}$; if those differences vanish, the sum is zero.

**Consequence for band-pass responses.** Let $B(\cdot)$ be any admissible band-pass kernel with $B(0) = 0$. If $\mathbf{x}$ is spatially constant in the sense of Lemma A.4, then $B(L)\mathbf{x} = \mathbf{0}$ and the captured energy

$$E = \frac{\|B(L)\mathbf{x}\|_2^2}{\|\mathbf{x}\|_2^2}$$

is also zero. Intuitively, spatially constant signals contain only the DC (zero-frequency) component, which admissible band-pass filters suppress.

**Implication for dynamic graphs (temporal-only changes).** If at *every* time $t$ the snapshot signal is spatially constant on each connected component (e.g., $\mathbf{x}^t = c_t \mathbf{1}$ for unnormalized $L$, or $D^{-1/2}\mathbf{x}^{(t)} = c_t \mathbf{1}$ for $L_{\text{sym}}$), then

$$S_{\text{high}}^t = 0 \quad \text{and} \quad E_t = 0 \quad \text{for all } t,$$

even if the constants $(c_t)$ change abruptly over time. Thus, purely temporal changes that remain spatially smooth within each snapshot are invisible to snapshot spectral criteria by construction.

### A.2.2 LIMITATIONS OF DIRECTLY PORTING THE STATIC FREQUENCY-DOMAIN RECIPE TO DYNAMIC GRAPHS

The frequency-domain framework is well validated on static graphs. When it is transplanted to dynamic graphs via per-snapshot processing, the introduction of time brings two limitations that merit attention and should be addressed in modeling.

**1. Snapshot blind spot for purely temporal anomalies.** Let $L_t$ and $\mathbf{x}_t$ denote the Laplacian and node features at time $t$. If within each snapshot the signal is spatially smooth on every connected component (e.g., $\mathbf{x}_t = c_t \mathbf{1}$ component-wise), then the high-frequency score

$$S_{\text{high}}^t \;=\; \frac{(\mathbf{x}_t)^\top L_t \mathbf{x}_t}{\|\mathbf{x}_t\|_2^2}$$

is 0 for all $t$ because $L_t \mathbf{1} = 0$. Hence, even if there is an abrupt temporal change $c_t \neq c_{t-1}$, snapshot-only spectral criteria do not explicitly reflect such anomalies that vary over time while remaining spatially smooth within each snapshot.

**2. Band mis-centering under nonstationary spectral "right-shift".** Let $\mu_t$ be the spectral measure of $\mathbf{x}_t$ with respect to $L_t$, and define the energy captured by a fixed band-pass $B$ as

$$E_t \;=\; \int B(\lambda)^2 \, d\mu_t(\lambda).$$

Assume that at time $t$ at least a fraction $\rho_t$ of the mass lies in $I_t = [\omega_t - \epsilon, \; \omega_t + \epsilon]$ (the empirical high-frequency center $\omega_t$). If $B$ is centered at $\omega_\star$ with half-bandwidth $b$ and Lipschitz constant $L_B$, then whenever $|\omega_t - \omega_\star| > b + \epsilon$,

$$E_t \;\leq\; (1 - \rho_t)\|B\|_\infty^2 \;+\; \rho_t \left( B(\omega_\star) - L_B \big( |\omega_t - \omega_\star| - b - \epsilon \big) \right)^2.$$

i.e., once the center drift exceeds the passband, the captured anomaly energy degrades markedly. As $\{\omega_t\}$ drifts over time, $\{E_t\}$ fluctuates across snapshots, revealing a systematic misalignment caused by temporal nonstationarity; cross-snapshot information is therefore needed to align and complement snapshot cues.

### A.3 LIMITATIONS IN THE PU SCENARIO

In this section, we provide a concise theoretical analysis of standard learning strategies under the Positive-Unlabeled (PU) setting, focusing on the widely used **naive approach**. We formalize its risk estimation and demonstrate its inherent bias, which leads to model collapse—a key failure mode for GNN-based anomaly detection in dynamic graphs.

### A.3.1 THE NAIVE APPROACH: FORMULATION AND BIAS

In PU learning, the absence of labeled negatives often leads practitioners to treat all unlabeled nodes $V^U$ as negatives. Let $V^L$ denote the set of labeled positives, and $f$ be the anomaly scoring function with predictions $\hat{y}_v = f(\mathbf{z}_v) \in [0, 1]$. The empirical risk for the naive approach is:

$$\hat{R}_{\text{naive}}(f) = \hat{\pi}_P \left( 1 - \frac{1}{|V^L|} \sum_{v \in V^L} \hat{y}_v \right) + (1 - \hat{\pi}_P) \left( \frac{1}{|V^U|} \sum_{v \in V^U} \hat{y}_v \right), \tag{21}$$

where $\hat{\pi}_P = \frac{|V^L|}{|V^L| + |V^U|}$ is the observed positive label ratio, typically much smaller than the true class prior $\pi_P$.

The flaw lies in the second term: it assumes the mean prediction over $V^U$ estimates the negative risk, i.e., $\mathbb{E}_U[\hat{y}] \approx \mathbb{E}_N[\hat{y}]$. However, since $V^U$ contains both true negatives and hidden positives, this is incorrect. The bias is:

$$\text{Bias} = \mathbb{E}_U[\hat{y}] - \mathbb{E}_N[\hat{y}] = \pi_P \left( \mathbb{E}_P[\hat{y}] - \mathbb{E}_N[\hat{y}] \right) \neq 0,$$

making $\hat{R}_{\text{naive}}(f)$ a biased estimator of the true risk $R(f)$ (Du Plessis et al., 2014; Bekker & Davis, 2020).

### A.3.2 MODEL COLLAPSE DRIVEN BY BIAS

This bias distorts optimization. In practice, $|V^U| \gg |V^L|$, so $(1 - \hat{\pi}_P) \approx 1$ and the second term in equation 21 dominates the loss. Minimizing $\hat{R}_{\text{naive}}(f)$ thus encourages the model to drive $\frac{1}{|V^U|} \sum_{v \in V^U} \hat{y}_v \to 0$, i.e., to predict all unlabeled nodes as normal ($\hat{y}_v = 0$), including hidden anomalies. This leads to *model collapse*: the classifier $f$ becomes biased toward the majority (normal) class and loses discriminative power for anomalies.

**Mathematical Illustration: Descent Toward Zero Predictor**    Let $\ell(a, y)$ be a differentiable loss (e.g., logistic or squared), and consider the population risk:

$$R_{\text{naive}}(f) = \hat{\pi}_P \, \mathbb{E}_{v \sim P}[\ell(1, \hat{y}_v)] + (1 - \hat{\pi}_P) \, \mathbb{E}_{v \sim U}[\ell(0, \hat{y}_v)].$$

For a perturbation $\delta \hat{y}_v = -\epsilon$ for $v \in U$ (and 0 otherwise), the directional derivative is

$$DR_{\text{naive}}(f)[\delta] = (1 - \hat{\pi}_P) \, \mathbb{E}_{v \sim U} \left[ \partial_2 \ell(0, \hat{y}_v) \, (-\epsilon) \right] < 0,$$

since $\partial_2 \ell(0, y) > 0$ for $y \in (0, 1]$. Thus, the loss always decreases by reducing $\hat{y}_v$ for $v \in U$, regardless of the true class.

For logistic loss, $\ell(a, y) = -a \log y - (1 - a) \log(1 - y)$, so $\partial_2 \ell(0, y) = \frac{1}{1-y} > 0$, yielding

$$DR_{\text{naive}}(f)[\delta] = -(1 - \hat{\pi}_P) \, \epsilon \, \mathbb{E}_U \left[ \frac{1}{1 - \hat{y}_v} \right] < 0.$$

This confirms a persistent descent direction toward the zero predictor, independent of the data's true structure.

### A.3.3 COUPLED AMPLIFICATION OF OVER-SMOOTHING AND PU BIAS

In the Positive–Unlabeled (PU) setting, the failure of GNN-based anomaly detection arises from a *coupled amplification* between architectural over-smoothing and supervision bias. On the one hand, during the forward pass, the message-passing encoder behaves as a low-pass filter: for node representations $\mathbf{Z}^{(k)} \in \mathbb{R}^{|V| \times d}$ and Laplacian $L$, the Dirichlet energy monotonically decreases,

$$\text{Tr}\left( (\mathbf{Z}^{(k)})^\top L \, \mathbf{Z}^{(k)} \right) \leq \text{Tr}\left( (\mathbf{Z}^{(k-1)})^\top L \, \mathbf{Z}^{(k-1)} \right),$$

which implies that high-frequency components carrying anomaly cues are exponentially attenuated across layers. On the other hand, *in the PU setting* with standard convex classification losses (e.g., logistic) and a dominant unlabeled set $U$, gradient-based optimization consistently decreases the unlabeled average prediction

$$\bar{y}_t^U := \frac{1}{|U|} \sum_{v \in U} \hat{y}_v^{(t)},$$

thereby pushing hidden positives toward zero. Crucially, the only corrective force that could increase $\bar{y}_t^U$—the representation-level separability of hidden positives in the high-frequency spectrum—is itself suppressed by the encoder's low-pass bias. Formally, one obtains a net drift inequality

$$\bar{y}_{t+1}^U - \bar{y}_t^U \leq -c_1 + c_2 \, \gamma^K,$$

where $c_1 > 0$ quantifies the persistent downward pull on $U$ under the PU setting, $c_2 \gamma^K$ upper-bounds the attenuated corrective signal after $K$ low-pass layers, and $\gamma < 1$ is the high-frequency attenuation factor. Whenever $c_1 > c_2 \gamma^K$, the sequence $\{\bar{y}_t^U\}$ decreases monotonically and approaches collapse. Hence the architectural bias (mechanism) and the supervision bias (incentive) are aligned, forming a self-reinforcing loop that annihilates anomaly signals—explaining why standard GNNs are particularly prone to failure in PU anomaly detection.

## B  THEORETICAL GUARANTEES OF THE WAVEN-PULL FRAMEWORK

### B.1  DYNAMIC GRAPH ENCODING

#### B.1.1  DEFINITIONS AND SETUP

**Graph Formulation.**  Let $\mathcal{G} = \{G_t\}_{t=1}^T$ be a dynamic graph, where $G_t = (V_t, A_t, X_t)$ is the graph snapshot at time $t$ with $|V| = n$ nodes, adjacency matrix $A_t$, and node feature matrix $X_t$. Let $L_t$ be the symmetrically normalized Laplacian of $G_t$ with eigendecomposition:

$$L_t = U_t \Lambda_t (U_t)^\top,$$

where $\Lambda_t = \mathrm{diag}(\lambda_1^t, \dots, \lambda_n^t)$ with $0 = \lambda_1^t \leq \dots \leq \lambda_n^t \leq 2$, and $U^t = [\mathbf{u}_1^t, \dots, \mathbf{u}_n^t]$ is the matrix of orthonormal eigenvectors.

**Beta-Wavelet Spectral Filter.**  Fix an order $C \in \mathbb{N}$. For any non-negative integers $p, q$ such that $p + q = C$, we define a Beta-wavelet spectral kernel:

$$g_{p,q}(\lambda) = \frac{1}{2B(p+1, q+1)} \left(\frac{\lambda}{2}\right)^p \left(1 - \frac{\lambda}{2}\right)^q, \quad \lambda \in [0, 2],$$

where $B(\cdot, \cdot)$ is the Beta function. The corresponding graph filter operator at time $t$ is defined as:

$$\mathcal{W}_{p,q}^t := g_{p,q}(L_t) = \frac{1}{2B(p+1, q+1)} \left(\frac{L_t}{2}\right)^p \left(I - \frac{L_t}{2}\right)^q.$$

This operator acts on a graph signal $\mathbf{x} \in \mathbb{R}^n$ by spectral multiplication: if $\hat{\mathbf{x}}_k = (\mathbf{u}_k^{(t)})^\top \mathbf{x}$, then the $k$-th spectral component of the filtered signal is $g_{p,q}(\lambda_k^{(t)}) \hat{\mathbf{x}}_k$.

**Temporal Attention Mechanism.**  Let $\mathbf{H}_t^{\mathrm{sp}}$ be the spatial node embeddings obtained after the Beta-wavelet graph convolution. The node-level temporal attention computes the representation for node $i$ at time $t$ as:

$$\mathbf{H}_t(i, :) = \mathbf{H}_t^{\mathrm{sp}}(i, :) + \sum_{k=1}^{t-1} \alpha_{i,k}^t \, \mathbf{H}_k^{\mathrm{sp}}(i, :),$$

where the attention weights $\alpha_{i,k}^t$ are computed using query and key projections. The output of the attention mechanism, representing the pure temporal context, is defined as the temporal embedding at time $t$:

$$\mathbf{H}_t^{\mathrm{att}} = \mathcal{A}_t(\{\mathbf{H}_k^{\mathrm{sp}}\}_{k=1}^{t-1}) = \sum_{k=1}^{t-1} \alpha_{i,k}^t \, \mathbf{H}_k^{\mathrm{sp}}(i, :).$$

The final node representation at time $t$ is then the sum of spatial and temporal embeddings:

$$\mathbf{H}_t = \mathbf{H}_t^{\mathrm{sp}} + \mathbf{H}_t^{\mathrm{att}}.$$

This decomposition allows us to separately analyze the effects of spatial filtering and temporal aggregation.

**Over-Smoothing Index.**  Fix $0 < \varepsilon < \eta \leq 2$. Define spectral projectors at time $t$:

$$P_t^{\mathrm{L}} := U_t \, \mathrm{diag}(\mathbf{1}\{\lambda_{t,k} \leq \varepsilon\}) \, U_t^\top, \qquad P_t^{\mathrm{H}} := U_t \, \mathrm{diag}(\mathbf{1}\{\lambda_{t,k} \geq \eta\}) \, U_t^\top.$$

For a vector $\mathbf{x} \in \mathbb{R}^{n_t}$, define the spectral contrast (over-smoothing index):

$$\mathrm{OS}_t(\mathbf{x}) := \frac{\|P_t^{\mathrm{L}} \mathbf{x}\|_2}{\|P_t^{\mathrm{H}} \mathbf{x}\|_2} \quad (\text{with } 0/0 := 0, \ a/0 := +\infty \text{ for } a > 0).$$

### B.1.2 CORE SPECTRAL PROPERTIES

**Lemma B.1** (Spectral properties of the Beta-wavelet filter). *For any $p \geq 1$, $q \geq 0$ with $p + q = C$, the kernel $g_{p,q}(\lambda)$ satisfies $g_{p,q}(0) = 0$. Furthermore, for any $0 < \varepsilon < \eta \leq 2$, there exist positive constants $c_1, c_2$ (depending on $p, q$) such that:*

$$\sup_{\lambda \in [0,\varepsilon]} g_{p,q}(\lambda) \leq c_1 \varepsilon^p, \qquad \inf_{\lambda \in [\eta,2]} g_{p,q}(\lambda) \geq c_2 \eta^p (1 - \eta/2)^q.$$

*In particular, for $p \geq 1$, $g_{p,q}$ is a strict high-pass or band-pass filter.*

*Proof.* Since $p \geq 1$, $g_{p,q}(\lambda) \sim \lambda^p$ as $\lambda \downarrow 0$, hence $g_{p,q}(0) = 0$ and $\sup_{\lambda \in [0,\varepsilon]} g_{p,q}(\lambda) \leq c_1' \varepsilon^p$ for some constant $c_1' > 0$. For the lower bound, $g_{p,q}$ is continuous and positive on the compact set $[\eta, 2]$, hence achieves a positive minimum there. This minimum can be bounded below by $c_2' \eta^p (1 - \eta/2)^q$ for some constant $c_2' > 0$. The constants $c_1$ and $c_2$ in the statement are precisely $c_1'$ and $c_2'$. $\square$

### B.1.3 PROOF OF SPECTRAL-CONTRAST CONTRACTION

**Proposition B.2** (Spectral-contrast contraction). *For any $\mathbf{x} \in \mathbb{R}^{n_t}$, time $t$, and filter $g_{p,q}$ with $p \geq 1$,*

$$\mathrm{OS}_t(\mathcal{W}_{p,q}^t \mathbf{x}) \leq \kappa_{p,q}(\varepsilon, \eta) \, \mathrm{OS}_t(\mathbf{x}), \quad \text{where} \quad \kappa_{p,q}(\varepsilon, \eta) := \frac{\sup_{\lambda \in [0,\varepsilon]} g_{p,q}(\lambda)}{\inf_{\lambda \in [\eta,2]} g_{p,q}(\lambda)}.$$

*Furthermore, from Lemma B.1, it follows that*

$$\kappa_{p,q}(\varepsilon, \eta) \leq \frac{c_1}{c_2} \cdot \frac{\varepsilon^p}{\eta^p (1 - \eta/2)^q} < 1,$$

*where $c_1, c_2$ are the positive constants from Lemma B.1.*

*Proof.* Let $\mathbf{y} = \mathcal{W}_{p,q}^t \mathbf{x}$. Recall that $\mathbf{y} = g_{p,q}(L_t)\mathbf{x}$. Any graph signal $\mathbf{x}$ admits a spectral representation $\mathbf{x} = \sum_k \hat{\mathbf{x}}_k \mathbf{u}_{t,k}$ with $\hat{\mathbf{x}}_k = \mathbf{u}_{t,k}^\top \mathbf{x}$. Thus, the filtered signal is $\mathbf{y} = \sum_k g_{p,q}(\lambda_{t,k}) \hat{\mathbf{x}}_k \mathbf{u}_{t,k}$, and its GFT coefficients are $\hat{\mathbf{y}}_k = g_{p,q}(\lambda_{t,k}) \hat{\mathbf{x}}_k$.

The energy of the low-frequency projection of $\mathbf{y}$ is:

$$\|P_t^{\mathrm{L}} \mathbf{y}\|_2^2 = \sum_{\lambda_{t,k} \leq \varepsilon} |\hat{\mathbf{y}}_k|^2 = \sum_{\lambda_{t,k} \leq \varepsilon} |g_{p,q}(\lambda_{t,k})|^2 |\hat{\mathbf{x}}_k|^2.$$

Since $|g_{p,q}(\lambda_{t,k})|^2 \leq \left(\sup_{\lambda \in [0,\varepsilon]} g_{p,q}(\lambda)\right)^2$ for all $\lambda_{t,k} \leq \varepsilon$, it follows that:

$$\|P_t^{\mathrm{L}} \mathbf{y}\|_2^2 \leq \left(\sup_{\lambda \in [0,\varepsilon]} g_{p,q}(\lambda)\right)^2 \sum_{\lambda_{t,k} \leq \varepsilon} |\hat{\mathbf{x}}_k|^2 = \left(\sup_{\lambda \in [0,\varepsilon]} g_{p,q}(\lambda)\right)^2 \|P_t^{\mathrm{L}} \mathbf{x}\|_2^2.$$

Similarly, the energy of the high-frequency projection of $\mathbf{y}$ is:

$$\|P_t^{\mathrm{H}} \mathbf{y}\|_2^2 = \sum_{\lambda_{t,k} \geq \eta} |\hat{\mathbf{y}}_k|^2 = \sum_{\lambda_{t,k} \geq \eta} |g_{p,q}(\lambda_{t,k})|^2 |\hat{\mathbf{x}}_k|^2.$$

Since $|g_{p,q}(\lambda_{t,k})|^2 \geq \left(\inf_{\lambda \in [\eta,2]} g_{p,q}(\lambda)\right)^2$ for all $\lambda_{t,k} \geq \eta$, it follows that:

$$\|P_t^{\mathrm{H}} \mathbf{y}\|_2^2 \geq \left(\inf_{\lambda \in [\eta,2]} g_{p,q}(\lambda)\right)^2 \sum_{\lambda_{t,k} \geq \eta} |\hat{\mathbf{x}}_k|^2 = \left(\inf_{\lambda \in [\eta,2]} g_{p,q}(\lambda)\right)^2 \|P_t^{\mathrm{H}} \mathbf{x}\|_2^2.$$

Taking the square root of both sides of the two inequalities yields:

$$\|P_t^{\mathrm{L}} \mathbf{y}\|_2 \leq \left(\sup_{\lambda \in [0,\varepsilon]} g_{p,q}(\lambda)\right) \|P_t^{\mathrm{L}} \mathbf{x}\|_2,$$

$$\|P_t^{\mathrm{H}}\mathbf{y}\|_2 \geq \left(\inf_{\lambda \in [\eta,2]} g_{p,q}(\lambda)\right) \|P_t^{\mathrm{H}}\mathbf{x}\|_2.$$

Finally, forming the ratio $\mathrm{OS}_t(\mathbf{y}) = \frac{\|P_t^{\mathrm{L}}\mathbf{y}\|_2}{\|P_t^{\mathrm{H}}\mathbf{y}\|_2}$ and combining the above inequalities gives the desired result:

$$\mathrm{OS}_t(\mathbf{y}) \leq \frac{\sup_{\lambda \in [0,\varepsilon]} g_{p,q}(\lambda)}{\inf_{\lambda \in [\eta,2]} g_{p,q}(\lambda)} \cdot \frac{\|P_t^{\mathrm{L}}\mathbf{x}\|_2}{\|P_t^{\mathrm{H}}\mathbf{x}\|_2} = \kappa_{p,q}(\varepsilon,\eta)\, \mathrm{OS}_t(\mathbf{x}).$$

The bound on $\kappa_{p,q}(\varepsilon,\eta)$ is obtained by substituting the inequalities from Lemma B.1 into its definition:

$$\kappa_{p,q}(\varepsilon,\eta) \leq \frac{c_1 \varepsilon^p}{c_2 \eta^p (1 - \eta/2)^q} = \frac{c_1}{c_2} \frac{(\varepsilon/\eta)^p}{(1 - \eta/2)^q}.$$

The term $(\varepsilon/\eta)^p < 1$ is the primary driver of the contraction, ensuring that $\kappa < 1$ under practical choices of spectral bands (i.e., $\eta$ not arbitrarily close to 2). This confirms that applying the filter necessarily reduces the over-smoothing index. $\qquad\square$

**Corollary B.3** (High-frequency anomaly sensitivity). *Let an anomaly indicator signal $\mathbf{x} \in \mathbb{R}^{n_t}$ satisfy $\|P_t^{\mathrm{H}}\mathbf{x}\|_2 > 0$. Then,*

$$\|P_t^{\mathrm{H}}(\mathcal{W}_{p,q}^t \mathbf{x})\|_2 \geq \gamma_{p,q}\|P_t^{\mathrm{H}}\mathbf{x}\|_2, \quad \text{with } \gamma_{p,q} := \inf_{\lambda \in [\eta,2]} g_{p,q}(\lambda) > 0,$$

$$\|P_t^{\mathrm{L}}(\mathcal{W}_{p,q}^t \mathbf{x})\|_2 \leq \beta_{p,q}\|P_t^{\mathrm{L}}\mathbf{x}\|_2, \quad \text{with } \beta_{p,q} := \sup_{\lambda \in [0,\varepsilon]} g_{p,q}(\lambda).$$

*Since $\kappa_{p,q} = \beta_{p,q}/\gamma_{p,q} < 1$, the high-to-low-frequency signal ratio increases after filtering.*

### B.1.4 PROPERTIES OF TEMPORAL OPERATIONS

**Lemma B.4** (Temporal operations preserve spectral contrast). *The temporal attention operation $\mathcal{A}_t$ and its output $\mathbf{H}_t^{\mathrm{att}}$ are block-diagonal across nodes. For any signal $\mathbf{x} \in \mathbb{R}^n$, the outputs $\mathbf{z}^{\mathrm{att}} = \mathbf{H}_t^{\mathrm{att}}\mathbf{x}$ and $\mathbf{z} = \mathbf{H}_t \mathbf{x}$ satisfy that each component $z_i^{\mathrm{att}}$ and $z_i$ depends only on $\{(\mathbf{H}_k^{\mathrm{sp}}\mathbf{x})_i\}_{k=1}^{t-1}$ and $(\mathbf{H}_t^{\mathrm{sp}}\mathbf{x})_i$ respectively. Consequently, $\mathcal{A}_t$, $\mathbf{H}_t^{\mathrm{att}}$, and $\mathbf{H}_t$ do not perform spatial smoothing and cannot increase $\mathrm{OS}_t(\mathbf{x})$.*

*Proof.* The temporal attention computes a weighted sum of previous features for each node independently. There is no interaction between different nodes $i$ and $j$ within this operation, making it commutative with any node-wise projection. Therefore, it cannot reduce the relative strength of high-frequency components. The final representation $\mathbf{H}^{(t)}$ is a sum of two block-diagonal components and thus also block-diagonal. $\qquad\square$

### B.1.5 MAIN THEOREM: MITIGATION OF OVER-SMOOTHING

**Theorem B.5** (Mitigation of over-smoothing). *Consider the computation of node representations at time $t$:*

$$\mathbf{H}_t^{\mathrm{sp}} = \sigma\left(\left[\mathcal{W}_{0,C}^t \mathbf{X}_t \mathbf{W}^{(0)} \| \cdots \| \mathcal{W}_{C,0}^t \mathbf{X}_t \mathbf{W}^{(C)}\right] \mathbf{W}_{mix}\right),$$

$$\mathbf{H}_t^{\mathrm{att}} = \mathcal{A}_t(\{\mathbf{H}_k^{\mathrm{sp}}\}_{k=1}^{t-1}),$$

$$\mathbf{H}_t = \mathbf{H}_t^{\mathrm{sp}} + \mathbf{H}_t^{\mathrm{att}}.$$

*Assume $\sigma$ is 1-Lipschitz and the linear transformations are arbitrary. Then for any nonzero $\mathbf{x} \in \mathbb{R}^{n_t}$:*

$$\mathrm{OS}_t(\mathbf{H}_t^{\mathrm{sp}}\mathbf{x}) \leq \max_{p \geq 1} \kappa_{p,C-p}(\varepsilon,\eta) \cdot \mathrm{OS}_t(\mathbf{X}_t \mathbf{x}),$$

$$\mathrm{OS}_t(\mathbf{H}_t^{\mathrm{att}}\mathbf{x}) \leq \max_{k \in \{1,\ldots,t-1\}} \mathrm{OS}_t(\mathbf{H}_k^{\mathrm{sp}}\mathbf{x}),$$

$$\mathrm{OS}_t(\mathbf{H}_t \mathbf{x}) \leq \max\left(\mathrm{OS}_t(\mathbf{H}_t^{\mathrm{sp}}\mathbf{x}), \mathrm{OS}_t(\mathbf{H}_t^{\mathrm{att}}\mathbf{x})\right).$$

*The composite block thus reduces over-smoothing while preserving temporal information.*

*Proof.* (Sketch) The first bound follows from Proposition B.2 applied to each filtered component ($p \geq 1$), noting that Lipschitz activation and linear mixing preserve the spectral contrast ratio. The second bound follows from Lemma B.4, as temporal attention does not introduce spatial mixing. The third bound follows from the linearity of spectral projectors and the fact that the norm of a sum is bounded by the sum of norms, and thus the ratio is dominated by the component with larger spectral contrast. □

*Remark* B.6 (Filter design for anomaly detection). Higher order $C$ provides richer spectral filters. Filters with $p \geq 1$ ensure strong low-frequency attenuation ($\sim \varepsilon^p$) while maintaining high-frequency response ($\sim \eta^p$), crucial for anomaly amplification. The parameter $q = C - p$ controls high-end attenuation ($\lambda \to 2$); $q \geq 1$ ensures numerical stability.

## B.2 THEORETICAL ANALYSIS OF THE PU-AWARE ALIGNMENT MODULE

This section provides a rigorous theoretical foundation for the PU-aware alignment module introduced in the main text. We structure our analysis into three parts. First, we derive our proposed label-alignment-aware objective from first principles. Second, we establish its theoretical guarantees, proving its equivalence to a constrained optimal classification problem. Finally, we analyze its robustness and stability in practical, dynamic settings.

### B.2.1 DERIVATION OF THE LABEL-ALIGNMENT-AWARE PU OBJECTIVE

We begin by tracing the derivation of our proposed objective, starting from the standard risk formulation for binary classification and adapting it to the Positive-Unlabeled (PU) setting.

**Setup.** Let $(X, Y) \sim p(x, y)$ with $Y \in \{0, 1\}$, $\pi_P := \mathbb{P}(Y = 1)$, $\pi_N = 1 - \pi_P$, class conditionals $p_P(x) := p(x \mid Y = 1)$ and $p_N(x) := p(x \mid Y = 0)$, and the unlabeled mixture $p_U(x) := \pi_P p_P(x) + \pi_N p_N(x)$. A probabilistic classifier is $f : \mathcal{X} \to \mathbb{R}$, and we define the predicted score as $\hat{y} = \sigma(f(x)) \in [0, 1]$, where $\sigma$ is the sigmoid function. In standard binary classification, a classifier $f$ minimizes the expected risk:

$$R(f) = \mathbb{E}_{(\boldsymbol{x}, y) \sim p(\boldsymbol{x}, y)} \left[ \mathbb{1} \left[ \hat{y} \neq y \right] \right],$$

where $\hat{y} = \mathbb{1}[f(x) \geq 0]$ is the predicted label. This risk can be decomposed into contributions from positive and negative samples:

$$\begin{aligned}
R(f) &= \pi_P \mathbb{E}_{\boldsymbol{x} \sim p_P(\boldsymbol{x})} \left[ \mathbb{1} \left[ \hat{y} \neq 1 \right] \right] + \pi_N \mathbb{E}_{\boldsymbol{x} \sim p_N(\boldsymbol{x})} \left[ \mathbb{1} \left[ \hat{y} \neq 0 \right] \right] \\
&= \pi_P \mathbb{E}_{\boldsymbol{x} \sim p_P(\boldsymbol{x})} \left[ 1 - \hat{y} \right] + \pi_N \mathbb{E}_{\boldsymbol{x} \sim p_N(\boldsymbol{x})} \left[ \hat{y} \right] \\
&= \pi_P \underbrace{\left| \mathbb{E}_{\boldsymbol{x} \sim p_P(\boldsymbol{x})} \left[ \hat{y} \right] - 1 \right|}_{R_P} + \pi_N \underbrace{\left| \mathbb{E}_{\boldsymbol{x} \sim p_N(\boldsymbol{x})} \left[ \hat{y} \right] - 0 \right|}_{R_N},
\end{aligned} \quad (22)$$

where $\pi_P$ and $\pi_N = 1 - \pi_P$ are the class priors, and $p_P(\boldsymbol{x})$, $p_N(\boldsymbol{x})$ are the class-conditional distributions. The expected risk thus measures the deviation between predicted and true label distributions; minimizing it aligns the predicted class proportions with the true ones Zhao et al. (2022).

**Lemma B.7** (PU decomposition and elimination of negatives). *For unlabeled data, which mixes positives and negatives, the expected ground-truth label equals the class prior:* $\mathbb{E}_{(\boldsymbol{x}, y) \sim p(\boldsymbol{x}, y)}[y] = \pi_P$. *With $p_U(\boldsymbol{x})$ as the unlabeled data distribution (a mixture of $p_P$ and $p_N$), the expected prediction is:*

$$\mathbb{E}_{\boldsymbol{x} \sim p_U(\boldsymbol{x})}[\hat{y}] = \pi_P \mathbb{E}_{\boldsymbol{x} \sim p_P(\boldsymbol{x})}[\hat{y}] + \pi_N \mathbb{E}_{\boldsymbol{x} \sim p_N(\boldsymbol{x})}[\hat{y}].$$

*Based on this decomposition, the negative risk $R_N$ can be indirectly estimated without access to negative labels.*

*Proof.*

$$\begin{aligned}
\pi_N R_N &= \mathbb{E}_{\boldsymbol{x} \sim p_U(\boldsymbol{x})} \left[ \hat{y} \right] - \pi_P \mathbb{E}_{\boldsymbol{x} \sim p_P(\boldsymbol{x})} \left[ \hat{y} \right] \\
&= \mathbb{E}_{\boldsymbol{x} \sim p_U(\boldsymbol{x})} \left[ \hat{y} \right] - \pi_P + \left( \pi_P - \pi_P \mathbb{E}_{\boldsymbol{x} \sim p_P(\boldsymbol{x})} \left[ \hat{y} \right] \right) \\
&= \mathbb{E}_{\boldsymbol{x} \sim p_U(\boldsymbol{x})} \left[ \hat{y} \right] - \pi_P + \pi_P R_P.
\end{aligned} \quad (23)$$

Substituting equation 23 into equation 22 yields an alternative expression for $R(f)$ that eliminates the need for explicit negative label computations, as follows:

$$R(f) = 2\pi_P R_P + \mathbb{E}_{\boldsymbol{x} \sim p_U(\boldsymbol{x})} \left[ \hat{y} \right] - \pi_P$$

To enforce alignment between the predicted label distribution on unlabeled data and the class prior, we add an absolute difference term to the risk function. This yields the label-alignment-aware risk $R(f)$:

$$R(f) = 2\pi_P R_P + \underbrace{\left| \mathbb{E}_{\boldsymbol{x} \sim p_U(\boldsymbol{x})} \left[ \hat{y} \right] - \pi_P \right|}_{R_U}. \quad (24)$$

$\square$

**Label-alignment-aware PU objective.** Motivated by equation 24, WAVEN-PULL uses the following PU objective, which is implemented and stable in practice:

$$R(f) \;=\; 2\pi_P(1 - \mathbb{E}_P[\hat{y}]) \;+\; |\mathbb{E}_U[\hat{y}] - \pi_P|. \tag{25}$$

This objective simultaneously maximizes the recall on labeled positives (minimizing $1 - \mathbb{E}_P[\hat{y}]$) and enforces alignment of the predicted positive rate on $U$ with the class prior $\pi_P$.

*Remark* B.8 (Interpretation of the Objective Components). The proposed objective consists of two core components addressing two distinct goals.

1. **Recall Maximization:** The term $2\pi_P(1 - \mathbb{E}_P[\hat{y}])$ serves as a continuous surrogate for maximizing recall. Minimizing this term is equivalent to maximizing the expected score on known positives, $\mathbb{E}_P[\hat{y}]$, which directly incentivizes the model to identify all true positive instances.

2. **Alignment Regularization:** The term $|\mathbb{E}_U[\hat{y}] - \pi_P|$ is a regularizer that enforces consistency with the class prior, preventing the model collapse associated with the naive PU objective.

### B.2.2 THEORETICAL GUARANTEES OF THE ALIGNMENT OBJECTIVE

Having derived our objective, we now provide its rigorous theoretical justification. We proceed in two steps. First, we show that our practical, unconstrained objective is an *exact* method for solving a more ideal constrained optimization problem. Second, we characterize the solution to this ideal problem and show that it is, in fact, the theoretically optimal classifier.

**Constrained alignment counterpart.** We also consider the constrained formulation corresponding to equation 25:

$$\min_f \; 2\pi_P\big(1 - \mathbb{E}_P[\hat{y}]\big) \quad \text{s.t.} \quad \mathbb{E}_U[\hat{y}] = \pi_P. \tag{26}$$

**Exact $L^1$-penalty at unit weight.** Throughout this part we take $\hat{y} = \sigma(f(x)) \in [0, 1]$ as the training-time score (hard decisions are a limiting case). To provide a rigorous theoretical grounding for our proposed objective in equation 25, we now establish its connection to the constrained optimization problem defined in equation 26. Specifically, we aim to show that our unconstrained, penalty-based objective is not merely an approximation, but is in fact an *exact* method for solving the ideal constrained problem. The following proposition formalizes this equivalence under a mild condition.

**Proposition B.9** (Equivalence to a Constrained Problem). *Let $J^\star := 2\pi_P\big(1 - \mathbb{E}_P[\hat{y}^\star]\big)$ be the optimal value of the constrained problem equation 26 at some feasible optimizer $f^\star$. Assume the KKT conditions hold at $f^\star$ with Lagrange multiplier $\nu^\star \in \mathbb{R}$. If $|\nu^\star| \le 1$, then any global minimizer of equation 25 is feasible for equation 26 and also minimizes equation 26. In other words, the unit $L^1$ penalty in equation 25 is* exact *under the bounded-multiplier condition.*

*Proof sketch.* Using $|a| = \max_{|\nu| \le 1} \nu a$, we rewrite equation 25 as

$$R(f) = \max_{|\nu| \le 1} \Big\{ 2\pi_P\big(1 - \mathbb{E}_P[\hat{y}]\big) + \nu\big(\mathbb{E}_U[\hat{y}] - \pi_P\big) \Big\} =: \max_{|\nu| \le 1} L(f, \nu).$$

Let $\nu^\star$ be a KKT multiplier for equation 26 at $f^\star$. If $|\nu^\star| \le 1$, then $(f^\star, \nu^\star)$ is a saddle point on $\{|\nu| \le 1\}$: for any $\nu$, $L(f^\star, \nu) \le L(f^\star, \nu^\star)$ (feasibility gives $\mathbb{E}_U[\hat{y}^\star] = \pi_P$); for any $f$, $L(f, \nu^\star) \ge L(f^\star, \nu^\star)$ (KKT optimality). Thus

$$\min_f R(f) = \min_f \max_{|\nu| \le 1} L(f, \nu) \;\ge\; \max_{|\nu| \le 1} \min_f L(f, \nu) \;\ge\; L(f^\star, \nu^\star) = J^\star,$$

while any feasible $f$ satisfies $R(f) = 2\pi_P(1 - \mathbb{E}_P[\hat{y}]) \ge J^\star$ with equality at $f^\star$. Hence every global minimizer of equation 25 is feasible and solves equation 26. $\square$

*Remark* B.10 (Near-exactness via worst-case slope). Define the worst-case slope (as in your original text)

$$\lambda_{\text{ex}} := \sup_{\mathbb{E}_U[\hat{y}] \ne \pi_P} \frac{2\pi_P\big(1 - \mathbb{E}_P[\hat{y}]\big) - J^\star}{\big|\mathbb{E}_U[\hat{y}] - \pi_P\big|} \in [0, \infty].$$

Then for any $f$,

$$R(f) \geq J^\star + \left(1 - \lambda_{\text{ex}}\right) \cdot \left|\mathbb{E}_U[\hat{y}] - \pi_P\right|.$$

Thus if $\lambda_{\text{ex}} < 1$, the unit-penalty objective $R(f)$ remains exact; if $\lambda_{\text{ex}} \geq 1$, the inequality quantifies the trade-off between suboptimality and alignment residual for analysis (and can be instantiated with the EMA-smoothed estimate in the dynamic setting).

The preceding analysis established that our PU objective is an exact method for the constrained problem. This naturally raises a question: What is the nature of the optimal solution to this problem? The following theorem leverages the classic Neyman-Pearson lemma to provide a precise answer.

**Theorem B.11** (Optimality of the Constrained Solution). *Among all measurable scorers $\hat{y} = \sigma(f(x)) \in [0,1]$ satisfying the alignment constraint $\mathbb{E}_U[\hat{y}] = \pi_P$, the maximum of $\mathbb{E}_P[\hat{y}]$ (equivalently, the minimum of $1 - \mathbb{E}_P[\hat{y}]$) is attained by a (possibly randomized) threshold on the likelihood ratio $L(x) := \frac{p_P(x)}{p_U(x)}$. In particular, there exist a threshold $\tau$ and a boundary randomization $\gamma(x) \in [0,1]$ such that*

$$\hat{y}^\star(x) = \mathbb{1}\{L(x) > \tau\} + \gamma(x)\,\mathbb{1}\{L(x) = \tau\} \quad \text{and} \quad \mathbb{E}_U[\hat{y}^\star] = \pi_P,$$

*and $\hat{y}^\star$ maximizes $\mathbb{E}_P[\hat{y}]$ among all feasible scorers. Equivalently, any maximizer is (almost surely) a nondecreasing transform of $L(x)$, or of the posterior $\eta_U(x) := \mathbb{P}(Y = 1 \mid X = x,\ X \sim p_U) = \frac{\pi_P\, p_P(x)}{p_U(x)}$.*

*Proof sketch.* This is the Neyman–Pearson lemma with the "size" measured under $p_U$. For any fixed $\alpha = \mathbb{E}_U[\hat{y}]$, the tests that maximize $\mathbb{E}_P[\hat{y}]$ are (a.s.) thresholds on $L(x)$ with boundary randomization to meet the constraint exactly. Setting $\alpha = \pi_P$ yields the claim; monotone transforms preserve the ranking. $\square$

**Corollary B.12** (Global optimality of alignment+ranking). *Under the exactness condition of Proposition B.9 (e.g., $|\nu^\star| \leq 1$ or $\lambda_{\text{ex}} < 1$), any global minimizer of the penalized objective equation 25 is feasible for the constrained problem equation 26. By Theorem B.11, its output score (almost surely) implements a likelihood-ratio threshold and thus induces the Neyman–Pearson optimal PU ranking under the alignment constraint.*

### B.3 GUARANTEES FOR PRACTICAL IMPLEMENTATION

Finally, we address two critical challenges of practical implementation: the temporal noise inherent in dynamic graphs and the uncertainty in the class prior $\pi_P$.

### B.3.1 TEMPORAL STABILIZATION USING EMA

In practice, our objective is optimized using empirical estimates from batches of data. However, in dynamic settings, the empirical estimate of $\mathbb{E}_U[\hat{y}]$ can exhibit high variance across timesteps, hindering stable training. To mitigate this, we employ an Exponential Moving Average (EMA) to smooth this estimate over time.

**Proper surrogate and scores.** In practice, WAVEN-PULL uses a score $f(x)$ parameterized by a neural network and optimizes the objective $R_{\text{final}}$ directly using empirical estimates and gradient descent. The empirical risk over a batch is:

$$\widehat{R}_{\text{final}}(f) = 2\pi_P \left(1 - \frac{1}{|V^L|} \sum_{v \in V^L} \hat{y}_v\right) + \left|\frac{1}{|V^U|} \sum_{v \in V^U} \hat{y}_v - \pi_P\right|.$$

Crucially, to mitigate the instability of the empirical estimate $\frac{1}{|V^U|} \sum \hat{y}_v$ in dynamic settings, WAVEN-PULL employs an Exponential Moving Average (EMA) to smooth this quantity over time, as described in Equation (13) of the main text. This replaces the second term with $|\hat{\mu}_t - \pi_P|$, where $\hat{\mu}_t$ is the EMA-smoothed estimate. This temporal stabilization is key to robust optimization in dynamic graphs.

**EMA stability: essential facts.** Let $e_t := \mathbb{E}[\hat{y}_U^{(t)}] - \pi_P$ and define the EMA

$$\hat{\mu}_t \;=\; \alpha_U \, \hat{\mu}_{t-1} + (1 - \alpha_U) \, \mathbb{E}[\hat{y}_U^{(t)}], \quad \hat{\mu}_1 = \mathbb{E}[\hat{y}_U^{(1)}], \quad \alpha_U \in [0, 1),$$

and set $\tilde{e}_t := \hat{\mu}_t - \pi_P$. Equivalently,

$$\tilde{e}_t \;=\; \alpha_U \tilde{e}_{t-1} + (1 - \alpha_U)e_t \;=\; (1 - \alpha_U) \sum_{k=0}^{t-1} \alpha_U^k \, e_{t-k}. \tag{27}$$

**Lemma B.13.** *Stability: Bound on Error Amplification] For all $t \geq 2$,*

$$|\tilde{e}_t| \;\leq\; \alpha_U |\tilde{e}_{t-1}| + (1 - \alpha_U)|e_t| \;\leq\; \max\{\, |\tilde{e}_{t-1}|, \; |e_t| \,\}.$$

*With $\tilde{e}_1 = e_1$, it follows that $|\tilde{e}_t| \leq \max_{1 \leq s \leq t} |e_s|$ for all $t$.*

*Proof.* Use $|\alpha a + (1 - \alpha)b| \leq \alpha|a| + (1 - \alpha)|b|$ with $\alpha = \alpha_U$, then the fact that a convex combination is bounded by the maximum of its endpoints. $\square$

**Lemma B.14** (Accuracy: Asymptotic Consistency). *If $e_t \to e_\infty$ as $t \to \infty$, then $\tilde{e}_t \to e_\infty$.*

*Proof.* Let $z_t := \tilde{e}_t - e_\infty$ and $u_t := e_t - e_\infty \to 0$. From equation 27, $z_t = \alpha_U z_{t-1} + (1 - \alpha_U)u_t$. Taking $\limsup$ of absolute values gives $\limsup_t |z_t| \leq \alpha_U \limsup_t |z_t|$, so $\limsup_t |z_t| = 0$ since $\alpha_U < 1$. $\square$

**Proposition B.15** (Effectiveness: Variance Reduction). *Assume $e_t = m + \xi_t$ where $(\xi_t)$ is zero-mean, uncorrelated across time with $\mathrm{Var}(\xi_t) = \sigma^2$, and independent of $\tilde{e}_{t-1}$. Then*

$$\mathrm{Var}(\tilde{e}_t) \;=\; \alpha_U^2 \, \mathrm{Var}(\tilde{e}_{t-1}) + (1 - \alpha_U)^2 \sigma^2,$$

*so $\mathrm{Var}(\tilde{e}_t)$ converges to the unique fixed point*

$$\mathrm{Var}_\infty(\tilde{e}) \;=\; \frac{(1 - \alpha_U)^2}{1 - \alpha_U^2} \, \sigma^2 \;=\; \frac{1 - \alpha_U}{1 + \alpha_U} \, \sigma^2 \;\leq\; \sigma^2,$$

*with strict reduction for any $\alpha_U \in (0, 1)$.*

### B.3.2   ROBUSTNESS TO PRIOR MISSPECIFICATION

In practice, the true class prior $\pi$ is unknown and must be estimated as $\hat{\pi}$. We now formally analyze the impact of small errors in this estimation, proving that our method is robust to such misspecifications. Specifically, we show that a small error in the prior leads to only a small, bounded change in the objective function, and that the resulting trained model remains near-optimal.

**Proposition B.16** (Robustness of the Objective Function). *Let $R_{final}^{(\pi)}(f) := 2\pi\bigl(1 - \mathbb{E}_P[\hat{y}]\bigr) + \bigl|\mathbb{E}_U[\hat{y}] - \pi\bigr|$. If one uses $\hat{\pi}$ in place of the true $\pi$, with $|\hat{\pi} - \pi| \leq \epsilon$, then for any $f$,*

$$\bigl| R_{final}^{(\hat{\pi})}(f) - R_{final}^{(\pi)}(f) \bigr| \;\leq\; 2\epsilon \;+\; \epsilon \;=\; 3\,\epsilon.$$

*Proof.* Write

$$R_{\mathrm{final}}^{(\hat{\pi})}(f) - R_{\mathrm{final}}^{(\pi)}(f) = 2(\hat{\pi} - \pi)\bigl(1 - \mathbb{E}_P[\hat{y}]\bigr) + \bigl(\, |\mathbb{E}_U[\hat{y}] - \hat{\pi}| - |\mathbb{E}_U[\hat{y}] - \pi| \,\bigr).$$

Take absolute values. Since $0 \leq 1 - \mathbb{E}_P[\hat{y}] \leq 1$, the first term is bounded by $2|\hat{\pi} - \pi| \leq 2\epsilon$. For the second term, use $||a| - |b|| \leq |a - b|$ with $a = \mathbb{E}_U[\hat{y}] - \hat{\pi}$, $b = \mathbb{E}_U[\hat{y}] - \pi$, giving a bound $|\hat{\pi} - \pi| \leq \epsilon$. Sum them to get $3\epsilon$. $\square$

**Corollary B.17** (Near-Optimality of the Learned Model). *Let $\delta := 3\epsilon$ as in Proposition B.16. Then*

$$\bigl| \min_f R_{final}^{(\hat{\pi})}(f) - \min_f R_{final}^{(\pi)}(f) \bigr| \;\leq\; \delta,$$

*and if $\hat{f} \in \arg\min_f R_{final}^{(\hat{\pi})}(f)$, then*

$$R_{final}^{(\pi)}(\hat{f}) \;\leq\; \min_f R_{final}^{(\pi)}(f) \;+\; 2\delta \;=\; \min_f R_{final}^{(\pi)}(f) \;+\; 6\epsilon.$$

*Remark* B.18 (Path-tightened bound and full-scale normalization). The bound in Proposition B.16 can be written more tightly along the optimization path as

$$\left| R_{\text{final}}^{(\hat{\pi})}(f) - R_{\text{final}}^{(\pi)}(f) \right| \leq 2\epsilon \left( 1 - \mathbb{E}_P[\hat{y}] \right) + \epsilon,$$

which is often much smaller than $3\epsilon$ as training improves recall (i.e., $\mathbb{E}_P[\hat{y}] \uparrow$). Moreover, since $R_{\text{final}}^{(\pi)}(f) \in [0,3]$, this implies a full-scale relative perturbation $\leq \epsilon$.

*Remark* B.19 (Transferring alignment under a misspecified prior). If training with the estimated prior achieves $|\mathbb{E}_U[\hat{y}] - \hat{\pi}| \leq \rho$, then by the triangle inequality

$$\left| \mathbb{E}_U[\hat{y}] - \pi \right| \leq \rho + \epsilon.$$

In particular, driving $|\mathbb{E}_U[\hat{y}] - \hat{\pi}| \to 0$ ensures the true alignment residual is $\leq \epsilon$.

### B.3.3 Proof of Proposition 4.1 (Theoretical Guarantees of Waven-Pull) in main text

Proposition 4.1 states that by jointly optimizing the dynamic graph encoder and the PU-aware alignment objective, WAVEN-PULL provably achieves three key properties. We prove each of these claims by synthesizing the results established throughout Appendix.

**(i) Proof of Preserving High-Frequency Structural Signals.**

This property is a direct consequence of the theoretical analysis of our dynamic graph encoder in Appendix B.1.

1. We formally quantify the degree of over-smoothing using the **Over-Smoothing Index**, defined as $\text{OS}_t(\mathbf{x}) := \|P_t^{\text{L}}\mathbf{x}\|_2 / \|P_t^{\text{H}}\mathbf{x}\|_2$. A smaller index indicates better preservation of high-frequency signals relative to low-frequency ones.

2. **Proposition B.2** proves that our Beta-wavelet spatial filter $\mathcal{W}_{p,q}^t$ acts as a **contraction map** on this index for any high-pass or band-pass channel ($p \geq 1$):

$$\text{OS}_t(\mathcal{W}_{p,q}^t \mathbf{x}) \leq \kappa_{p,q} \cdot \text{OS}_t(\mathbf{x}), \quad \text{where the contraction factor } \kappa_{p,q} < 1.$$

This mathematically guarantees that the spatial encoding step actively reduces over-smoothing.

3. Furthermore, **Lemma B.4** establishes that the temporal attention module does not perform spatial smoothing and therefore cannot increase the over-smoothing index.

4. Combining these results, **Theorem B.5** provides the overarching guarantee for the entire encoder, confirming that the final node representations $\mathbf{H}_t$ have a strictly controlled (and often reduced) over-smoothing index compared to the input features. This formally proves that the encoder preserves the high-frequency structural signals crucial for anomaly detection.

**(ii) Proof of Attaining an Optimal and Stable PU Classifier.**

This property is established by the rigorous analysis of our PU-aware alignment module in Appendix B.2.

1. **Optimality**: The analysis culminates in **Corollary B.12**, which synthesizes two key results. First, **Proposition B.9** proves that minimizing our practical, unconstrained objective Eq. equation 25 is mathematically **equivalent** to solving an ideal constrained problem Eq. equation 26. Second, **Theorem B.11** leverages the Neyman-Pearson lemma to prove that the solution to this ideal problem is the **theoretically optimal classifier** that maximizes recall under the alignment constraint. Therefore, our objective function guides the model towards a provably optimal solution.

2. **Stability**: In dynamic settings, stability is crucial. Our use of an Exponential Moving Average (EMA) is theoretically justified. **Lemma B.13** proves that the EMA is stable and **does not amplify temporal errors** ($|\tilde{e}_t| \leq \max_{s \leq t} |e_s|$). **Proposition B.15** further proves its effectiveness by showing it achieves a **strict variance reduction**:

$$\text{Var}_\infty(\tilde{e}) = \frac{1 - \alpha_U}{1 + \alpha_U} \sigma^2 < \sigma^2, \quad \text{for } \alpha_U \in (0,1).$$

These guarantees ensure that the classifier is not only optimal but also stable against the temporal noise inherent in dynamic graphs.

**(iii) Proof of Achieving Robustness Against Oversmoothing and Model Collapse.**

This overall robustness is a **joint consequence** of the encoder's properties (proven in (i)) and the objective's properties (proven previously).

1. **Robustness Against Over-smoothing**: As proven in (i), our encoder is designed to be an over-smoothing mitigator. It preserves the high-frequency signals that standard GNNs are proven to destroy (as analyzed in Appendix A). This provides the necessary **representational foundation** for the classifier to distinguish anomalies.

2. **Robustness Against Model Collapse**: As proven in (ii), our PU-aware objective is optimal and stable. The symmetric alignment term, $|\mathbb{E}_U[\hat{y}] - \pi_P|$, eliminates the degenerate descent direction that plagues naive PU objectives (analyzed in Appendix A.3). This provides the correct **unbiased supervision signal** that prevents the model from collapsing to a trivial solution.

3. **The Joint Effect**: The robustness of WAVEN-PULL arises from breaking the "coupled amplification" feedback loop described in Appendix A.3.3. The encoder **preserves** the anomaly signal (solving the representation problem), and the PU objective **incentivizes** the model to correctly use that signal (solving the supervision problem). Without the former, the latter has no signal to act upon. Without the latter, the preserved signal would be ignored due to bias. The joint optimization ensures both modules work in synergy, leading to a system that is robust to both over-smoothing and model collapse.

∎

## C  PRELIMINARY CONCEPTS

### C.1  DYNAMIC GRAPH.

A dynamic graph is a graph whose structure and node attributes evolve over time, reflecting changes in real-world systems. Formally, a dynamic graph is represented as a sequence of graph snapshots $\mathcal{G} = \{G^{(1)}, G^{(2)}, \ldots, G^{(T)}\}$, where each snapshot $G^{(t)} = (V^{(t)}, E^{(t)}, \mathbf{X}^{(t)})$ consists of a node set $V_t$, an edge set $E_t$, and a node feature matrix $\mathbf{X}^{(t)} \in \mathbb{R}^{|V_t| \times d}$ at time $t$. Both the set of nodes and edges, as well as their attributes, may change across time steps, resulting in varying graph sizes and structures. Dynamic graphs are prevalent in many domains, including social networks, financial transaction systems, and communication networks, where the relationships and behaviors of entities are inherently time-dependent. An example of a dynamic graph is shown in Figure 4.

Anomalies in such graphs often correspond to abrupt changes in connectivity patterns or unusual shifts in node attributes, making their detection particularly challenging. The temporal evolution of the graph introduces additional complexity, as the underlying patterns and distributions can shift, leading to instability in node behavior and making reliable prediction more difficult. Furthermore, in real-world applications, only a small fraction of anomalous nodes are typically labeled, due to the rarity of anomalies and the high cost of manual annotation. A key challenge in dynamic graph analysis is to effectively capture both spatial (structural) and temporal dependencies. Unlike static graphs, which provide only a single snapshot, dynamic graphs require models to leverage historical information and temporal context to uncover meaningful patterns and detect anomalies. Temporal dependencies arise from the evolution of node interactions and attribute distributions, which can influence the emergence and manifestation of anomalous behaviors. Therefore, robust anomaly detection in dynamic graphs demands learning representations that jointly encode both the structural and temporal aspects of the data.

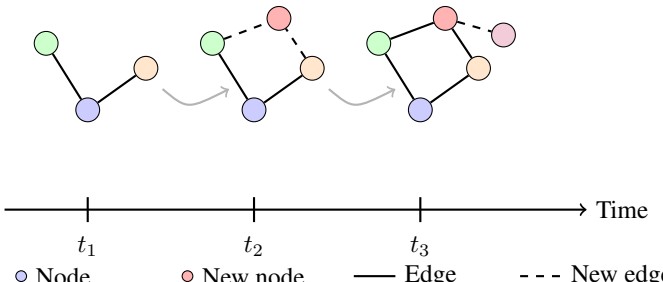

Figure 4: Illustration of a dynamic graph evolving over three time steps. Nodes and edges can appear or disappear as the graph evolves.

### C.2  POSITIVE-UNLABELED LEARNING

Positive-Unlabeled (PU) learning is a weakly supervised learning paradigm designed for scenarios where only a small subset of positive instances are labeled, while the vast majority of data remains unlabeled. Unlike traditional supervised learning, which requires both positive and negative labels, PU learning operates under the realistic constraint that negative labels are unavailable or prohibitively expensive to obtain. To effectively address this issue, PU learning commonly adopts the *Selected Completely At Random* (SCAR) assumption, which posits that labeled positives are randomly drawn from the true positive distribution, while unlabeled instances accurately represent the overall data distribution. This assumption is crucial as it ensures unbiased estimation of class probabilities from incomplete labels.

The PU learning problem is typically formulated within the empirical risk minimization (ERM) framework. Let $\pi_P = \Pr(y = 1)$ denote the class prior (i.e., the proportion of positives in the data). The expected classification risk can be expressed as:

$$R(f) = \pi_P \cdot \mathbb{E}_{x \sim p_P(x)}[\ell(f(x), 1)] + (1 - \pi_P) \cdot \mathbb{E}_{x \sim p_N(x)}[\ell(f(x), 0)],$$

where $p_P(x)$ and $p_N(x)$ are the distributions of positive and negative samples, respectively, and $\ell$ is a loss function. Since negative labels are not available, the second term is estimated indirectly

using the unlabeled data, allowing the model to be trained using only labeled positives and unlabeled examples Garg et al. (2021). This risk decomposition forms the basis for a variety of PU learning algorithms, such as unbiased PU (uPU) Du Plessis et al. (2014) and non-negative PU (nnPU) Kiryo et al. (2017).

In the context of dynamic graph anomaly detection, this PU setting naturally arises: only a handful of anomalous nodes are labeled, while the rest of the nodes are unlabeled and may include both normal and anomalous cases. Standard supervised approaches, which require explicit negative labels, are thus inapplicable. PU learning provides a principled and practical alternative by leveraging the available labeled anomalies and the abundant unlabeled data, without relying on confirmed negatives. This approach has demonstrated success in diverse applications such as disease diagnosis, fraud detection, and web mining. In this work, we leverage the PU learning framework to tackle the challenge of dynamic anomaly detection under sparse anomaly supervision.

## D ADDITIONAL EXPERIMENTAL SETTINGS

### D.1 DATASETS

In this subsection, we provide a brief introduction to the datasets used in the experiments. The statistics are shown in Table 4.

Table 4: Statistics of datasets

| Dataset | #Nodes | #Edges | #Anomalies |
|---------|--------|--------|------------|
| Wikipedia | 9,227 | 157,474 | 217 |
| Reddit | 10,984 | 672,447 | 366 |
| Bit-Alpha | 3,783 | 24,186 | 874 |
| Bit-OTC | 5,881 | 35,592 | 2,568 |

**(i) Wikipedia Kumar et al. (2019).** This dataset captures the temporal evolution of user interactions on Wikipedia, one of the largest collaborative knowledge bases. Each node represents a user, and each edge corresponds to an edit event where a user modifies a page. The dynamic graph is constructed by aggregating edit events into temporal snapshots, reflecting the evolving structure of user collaborations and interactions over time. The binary label for each user indicates whether the user has been banned for violating Wikipedia's community guidelines, with banned users considered anomalous. The dataset is challenging due to the sparsity of labeled anomalies and the complex, evolving patterns of user behavior.

**(ii) Reddit Kumar et al. (2019).** This dataset consists of user activity logs from a specific subreddit within the Reddit platform, a popular online discussion forum. Nodes represent users, and edges denote interactions such as posting, commenting, or replying within the subreddit. The temporal aspect is captured by dividing the event stream into discrete time windows, forming a sequence of dynamic graph snapshots. Each user is assigned a binary label indicating whether they have been banned from the subreddit, with banned users treated as anomalies. The dataset is characterized by highly dynamic user engagement, bursty activity patterns, and a small fraction of labeled anomalies, making it a realistic and challenging benchmark for dynamic anomaly detection.

**(iii) Bitcoin-Alpha (Bit-Alpha) Lee et al. (2024).** This dataset is derived from the Bitcoin Alpha trust network, an online platform where users trade Bitcoin and rate each other based on transaction experiences. Nodes correspond to users, and directed edges represent trust ratings, with each edge carrying a signed score from -10 (complete distrust) to +10 (complete trust). The dynamic graph is constructed by segmenting the rating events into temporal snapshots, capturing the evolution of trust relationships over time. Anomalous users are labeled following the preprocessing procedure of SLADE Lee et al. (2024), where node-level anomaly states are inferred from aggregated trust interactions rather than from individual edges. The dataset poses unique challenges due to the signed, weighted nature of edges, the presence of both positive and negative interactions, and the temporal drift in user behavior.

**(iv) Bitcoin-OTC (Bit-OTC) Lee et al. (2024).** Similar in structure to Bitcoin-Alpha, the Bitcoin-OTC dataset originates from the OTC (Over-The-Counter) Bitcoin trading platform, where users

assign trust ratings to one another following transactions. Each node is a user, and each directed edge encodes a trust score in the range of -10 to +10. The dynamic graph is formed by organizing trust rating events into time-based snapshots, enabling the study of evolving trust dynamics. Anomaly labels are obtained in the same way as in Bitcoin-Alpha, following the preprocessing procedure of SLADE Lee et al. (2024). The dataset is notable for its complex trust network, the rarity of labeled anomalies, and the need to distinguish between normal fluctuations in trust and genuine anomalous activity.

## D.2 BASELINES

Below, we summarize the baseline methods employed for comparison in our experiments:

**(i) TGAT Xu et al. (2020).** Temporal Graph Attention Network (TGAT) introduces a self-attention mechanism tailored for dynamic graphs, along with a novel time encoding scheme grounded in Bochner's theorem from harmonic analysis. This enables the model to effectively capture both structural and temporal dependencies in evolving graphs. In our experiments, TGAT is configured with two layers, two attention heads, and a dropout rate of 0.1 for optimal performance.

**(ii) TGN Rossi et al. (2020)** Temporal Graph Networks (TGN) provide a flexible framework for learning on continuous-time dynamic graphs. TGN incorporates a memory module to retain long-term node histories and employs embedding modules to address memory staleness. The architecture combines efficient parallel processing with attention-based graph aggregation, enabling scalable and expressive modeling of temporal interactions.

**(iii) GDN Ding et al. (2021).** Graph Deviation Network (GDN) leverages a small set of labeled anomalies to enforce statistically significant deviations between abnormal and normal nodes. The model learns generalizable knowledge that can be used for initialization and rapid adaptation to new nodes. In our setup, GDN is trained for 1000 epochs with a batch size of 16.

**(iv) AMNet Chai et al. (2022).** Adaptive Multi-frequency Network (AMNet) is designed for anomaly detection in attributed graphs by addressing the mismatch between GNNs' inherent low-pass filtering and the high-frequency nature of many anomalies. AMNet utilizes node-wise frequency filters parameterized by restricted Bernstein polynomials and a signal fusion module to effectively capture and integrate multi-frequency information for robust anomaly identification.

**(v) SAD Tian et al. (2023).** Self-supervised Anomaly Detection (SAD) combines a time-aware memory bank with a pseudo-label contrastive learning module to exploit large volumes of unlabeled data. The model adopts mini-batch training and samples two-hop subgraphs with 20 nodes per hop, enabling efficient and scalable anomaly detection in dynamic graphs.

**(vi) SLADE Lee et al. (2024).** SLADE is a self-supervised anomaly detection framework for dynamic edge streams. It employs temporal contrastive learning and memory generation tasks, leveraging TGAT-based aggregation and GRU-updated node memories. This design enables efficient, label-free detection of anomalies in evolving graph structures.

## D.3 EVALUATION METRICS

In this subsection, we provide a detailed explanation of the evaluation metrics used in our experiments.

**AUC.** The Area Under the Receiver Operating Characteristic Curve (**AUC**) is a widely used metric to evaluate the ranking quality of anomaly detection models. It measures the probability that a randomly chosen anomalous (positive) instance is ranked higher than a randomly chosen normal (negative) instance by the model. AUC is threshold-independent and provides a comprehensive assessment of the model's ability to distinguish between anomalies and normal nodes across all possible decision thresholds.

Formally, let $\mathcal{P}$ denote the set of positive (anomalous) samples and $\mathcal{N}$ denote the set of negative (normal) samples. Let $s(x)$ be the anomaly score assigned by the model to sample $x$. The AUC can be computed as:

$$\text{AUC} = \frac{1}{|\mathcal{P}| \cdot |\mathcal{N}|} \sum_{x_p \in \mathcal{P}} \sum_{x_n \in \mathcal{N}} \mathbb{I}\left(s(x_p) > s(x_n)\right),$$

where $\mathbb{I}(\cdot)$ is the indicator function, which equals 1 if the condition is true and 0 otherwise. An AUC of 1.0 indicates perfect separation between anomalies and normal nodes, while an AUC of 0.5 corresponds to random guessing.

**Precision@K.** Precision at $K$ (**Precision@K**) is a metric that quantifies the proportion of true anomalies among the top $K$ instances ranked by the model's anomaly scores. It reflects the accuracy of the model's highest-confidence predictions, which is particularly important in practical scenarios where only a limited number of top-ranked anomalies can be investigated.

Formally, let $\mathcal{S}_K$ denote the set of $K$ samples with the highest anomaly scores, and let $\mathcal{P}$ denote the set of ground-truth positive (anomalous) samples. Precision@K is defined as:

$$\text{Precision@}K = \frac{|\mathcal{S}_K \cap \mathcal{P}|}{K},$$

where $|\mathcal{S}_K \cap \mathcal{P}|$ is the number of true anomalies among the top $K$ predictions. A higher Precision@K indicates that the model is more effective at prioritizing true anomalies in its top-ranked outputs.

**Recall@K.** Recall at $K$ (**Recall@K**) is a metric that quantifies the proportion of true anomalies among the top $K$ instances ranked by the model's anomaly scores. It measures the coverage of the model's top-ranked predictions, indicating how many of the actual positive instances are included in the top $K$ ranked list.

Formally, let $\mathcal{S}_K$ denote the set of $K$ samples with the highest anomaly scores, and let $\mathcal{P}$ denote the set of ground-truth positive (anomalous) samples. Recall@K is defined as:

$$\text{Recall@}K = \frac{|\mathcal{S}_K \cap \mathcal{P}|}{|\mathcal{P}|},$$

where $|\mathcal{S}_K \cap \mathcal{P}|$ is the number of true anomalies among the top $K$ predictions. A higher Recall@K indicates that the model is more effective at capturing the majority of the actual positive instances in its top-ranked outputs.

## D.4 EXPERIMENTAL SETUP

The class prior $\pi_P$ was set to a default value of 0.01 for the main experiments. To reduce temporal fluctuations in predictions, we applied an exponential moving average (EMA) with a momentum parameter of $\alpha_U = 0.85$ to the statistics of unlabeled samples. For optimization, we employed the Adam optimizer Kingma & Ba (2014) with a constant learning rate of $1 \times 10^{-3}$ and a weight decay of $1 \times 10^{-4}$. The learning rate was further annealed using a cosine schedule without restarts. To simulate a realistic PU learning scenario, we fixed the number of labeled positive samples (num-labeled) at 100. During training, the batch size was set to 48 for positive samples ($P$), 128 for unlabeled samples ($U$), and 128 for testing. Each model was trained for 60 epochs. To ensure the robustness of our results, all experiments were repeated five times with different random seeds, and the average performance was reported. The experiments were executed using Python 3.8 on a server with an NVIDIA A40 GPU running CUDA 11.3.

## E ADDITIONAL EXPERIMENTAL ANALYSES

To further validate the effectiveness and reliability of WAVEN-PULL under diverse conditions, we present a comprehensive set of additional experimental analyses. These include: (1) a robustness analysis on key hyperparameters such as the number of snapshots, labeled anomaly batch size, and class prior; (2) a few-shot evaluation simulating deployment under minimal supervision on future time steps; and (3) an in-depth error analysis identifying factors contributing to false positive predictions. Together, these experiments demonstrate the stability, adaptability, and practical utility of our method in real-world PU-based dynamic anomaly detection scenarios.

### E.1 ROBUSTNESS ANALYSIS

In this section, we examine the robustness of WAVEN-PULL by evaluating its anomaly detection performance under various settings using AUC, Precision@50, and Recall@50 as representative metrics.

*Snapshot*. We vary the number of temporal slices (*snapshot*) from 4 to 8 to study its impact. As shown in Figure 5 (a), the best AUC on Wikipedia appears at $snapshot = 5$, while Reddit benefits from finer partitions, peaking at $snapshot = 6$. On the Bitcoin datasets, Bitcoinalpha achieves optimal performance at $snapshot = 7$, whereas Bitcoinotc prefers $snapshot = 6$. These results indicate that datasets with rapidly evolving anomalies favor finer-grained snapshots, while coarser segmentation suits more stable patterns. Overall, setting *snapshot* within 5–7 offers a good trade-off between detection accuracy and computational cost.

*P_batch_size*. This parameter controls the number of anomalous samples included in each training batch. We vary it from 32 to 64 to assess its impact. As shown in Figure 5 (b), setting $P\_batch\_size$ around 48 achieves the best AUC on Wikipedia, Bitcoinalpha, and Bitcoinotc, while Reddit slightly favors a smaller batch of 40 anomalies. Using too few anomalies (e.g., 32) may lead to unstable gradient updates and hinder the model's ability to learn meaningful anomaly patterns. Conversely, excessively large batches (e.g., 64) increase the risk of overfitting to the limited labeled anomalies, offering only marginal improvements or even degrading generalization. Overall, maintaining $P\_batch\_size$ between 40 and 56 achieves a robust balance between training stability and anomaly detection performance.

*Prior.* As discussed earlier, the ground-truth class prior $\pi_P$ is typically unavailable in real-world scenarios. Instead, our method adopts an adaptive, data-driven estimation strategy to approximate this prior during training Garg et al. (2021). To assess the robustness of our framework with respect to prior estimation error, we varied the value of $\pi_P$ from 0.001 to 0.5. As shown in Figure 5(c), WAVEN-PULL exhibits stable detection performance across a wide range of prior values. This demonstrates that the model is not overly sensitive to the exact value of $\pi_P$, and that our distribution alignment mechanism remains effective even when the prior is inaccurately specified. These results confirm the practicality of our prior estimation approach and the robustness of the training process under uncertain class prior conditions.

## E.2 ERROR ANALYSES

To better understand the limitations of our model, we conduct an error analysis focusing on false positives (FPs)—i.e., normal nodes that are incorrectly predicted as anomalies. Specifically, we analyze the top-$K$ nodes with the highest predicted anomaly scores and examine their structural and temporal properties. Representative examples of misclassified nodes are summarized in Table 5. We identify two prominent patterns contributing to these misclassifications:

- **Contamination by Abnormal Neighbors.** Some normal nodes are densely connected to multiple labeled anomalies, which may propagate misleading patterns. For instance, as shown in Table 5, node 7995 has 23 unique neighbors, including 8 anomalies. Similarly, node 6127 connects to 11 neighbors, 7 of whom are labeled anomalous. Such proximity to abnormal behavior can distort message aggregation and lead to overestimation of anomaly scores.

- **Abruptly Emerging Sparse Nodes.** Another category includes nodes that appear suddenly in the final snapshots of the dynamic graph, with very limited structural and temporal history. Examples like node 254, 108, and 2696 show minimal connectivity and activity—yet due to their abrupt emergence and lack of past context, the model may interpret their behavior as anomalous. This highlights a limitation of temporal models under extreme label sparsity, where lack of historical evidence makes newcomers harder to classify reliably.

These findings reveal two critical vulnerabilities of our model under extreme label sparsity: sensitivity to anomalous structural context and instability in classifying nodes with minimal historical data. These issues suggest two potential directions for further improvement. First, future work may explore neighborhood denoising or robust aggregation mechanisms to suppress the propagation of contaminated features from abnormal neighbors. Second, incorporating uncertainty-aware learning or memory-augmented modules could improve robustness in detecting anomalies among nodes with limited temporal history, such as abruptly emerging or sparsely connected nodes.

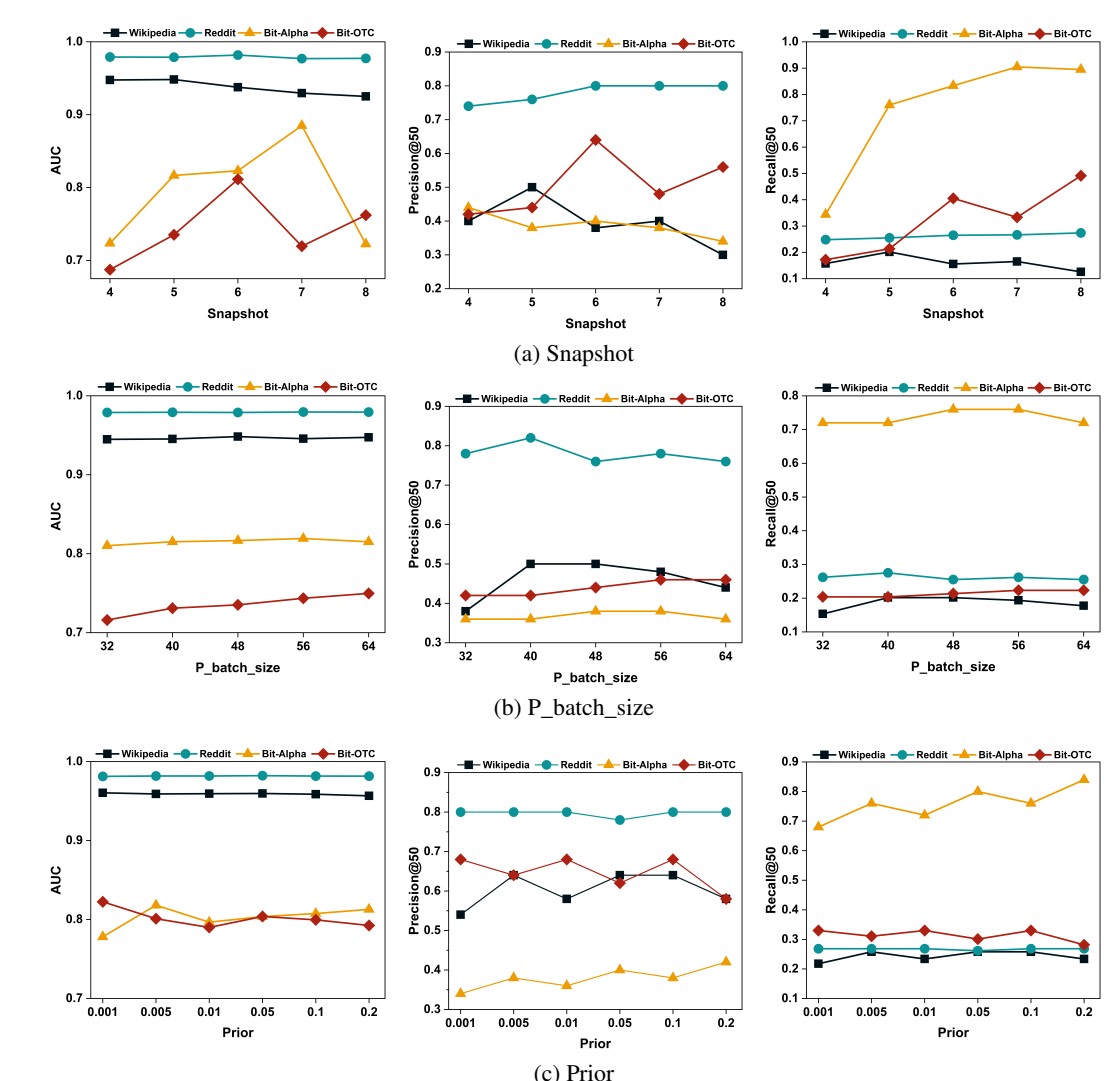

Figure 5: Performances analysis of WAVEN-PULL with various hyperparameter settings: (a) *Snapshot*, (b) *P_batch_size*, (c) *Prior*.

Table 5: Structural and Temporal Features of Selected False Positive Nodes

| Node ID | Total Edges | Unique Neighbors | Abnormal Neighbors | Active Snapshots |
|---------|-------------|------------------|--------------------|------------------|
| 644 | 145 | 17 | 5 | 1,2,3,4,5 |
| 2277 | 9 | 8 | 4 | 1,2,3,4,5 |
| 6127 | 30 | 11 | 7 | 1,2,3,4,5 |
| 7995 | 32 | 23 | 8 | 1,2,3,4,5 |
| 49 | 8 | 4 | 0 | 1 |
| 108 | 6 | 3 | 0 | 1 |
| 254 | 5 | 3 | 0 | 1 |
| 2696 | 2 | 1 | 0 | 2 |

### E.3 Few Shot Evaluation

To evaluate the generalization ability of WAVEN-PULL under limited supervision in future time steps, we design a few-shot evaluation protocol tailored to dynamic graph anomaly detection. Specifically, we treat the final snapshot in the dynamic graph sequence as a proxy for future data, while the preceding snapshots constitute historical observations. In our default experimental setup, 10% of the nodes in the final snapshot are included in the training set, from which 100 anomalous nodes are randomly labeled to simulate real-world settings where only a small number of anomalies are observed.

To further assess the model's capability in detecting anomalies under more challenging few-shot conditions, we vary the visibility of future data in the training set. In particular, we evaluate four scenarios where 10%, 5%, 1%, or 0% of the nodes from the final snapshot are included in training. This simulates practical deployment settings where few or no future anomalies are observable at training time. The model must therefore rely entirely on temporal patterns learned from historical data to identify anomalies in previously unseen nodes.

Table 6: Few-shot anomaly detection performance under varying proportions of future data observed during training.

| % Seen | Metric | Wikipedia | Reddit | Bit-Alpha | Bit-OTC |
|--------|--------|-----------|--------|-----------|---------|
| | AUC | 0.9483 | 0.9789 | 0.8166 | 0.7353 |
| 10% | Precision@50 | 0.3600 | 0.7600 | 0.3800 | 0.4400 |
| | Recall@50 | 0.1452 | 0.2550 | 0.7600 | 0.2136 |
| | AUC | 0.9479 | 0.9773 | 0.8142 | 0.7316 |
| 5% | Precision@50 | 0.3000 | 0.7600 | 0.3600 | 0.4600 |
| | Recall@50 | 0.1163 | 0.2389 | 0.7200 | 0.2129 |
| | AUC | 0.9466 | 0.9785 | 0.7835 | 0.7596 |
| 1% | Precision@50 | 0.2800 | 0.7800 | 0.3600 | 0.5600 |
| | Recall@50 | 0.1045 | 0.2393 | 0.6667 | 0.2545 |
| | AUC | 0.9440 | 0.9790 | 0.8261 | 0.7378 |
| 0% | Precision@50 | 0.3000 | 0.7600 | 0.4000 | 0.5800 |
| | Recall@50 | 0.1111 | 0.2331 | 0.7407 | 0.2589 |

We report the results of few-shot evaluation in Table 6, where different proportions (10%, 5%, 1%, and 0%) of nodes from the final snapshot are included in training. Each setting is evaluated using AUC, Precision@50, and Recall@50 across four datasets. Overall, the model maintains strong performance even with minimal future supervision. On Wikipedia and Reddit, AUC remains consistently above 0.94, though Precision@50 and Recall@50 show modest declines as fewer nodes are seen during training. Interestingly, on Bitcoin-Alpha and Bitcoin-OTC, the performance is robust and even improves slightly in the 0% case, suggesting that the model can effectively generalize to unseen future nodes based on historical representations. These results demonstrate that WAVEN-PULL is capable of detecting anomalies in future snapshots with very limited or no future supervision.

## F  USE OF LARGE LANGUAGE MODELS

We used Large Language Models (LLMs) to assist in polishing the manuscript. All content generated with the help of LLMs was carefully reviewed, verified, and edited by the authors to ensure accuracy and originality. We take full responsibility for all content in the paper, including any parts assisted by LLMs.

Table 7: Frequently-used symbols in this research

| Symbol | Meaning | Symbol | Meaning |
|---|---|---|---|
| $\mathcal{G}$ | Sequence of dynamic graph snapshots. | $L_t$ | Symmetric normalized Laplacian. |
| $V_t$ | Node set at time $t$. | $E_t$ | Edge set at time $t$. |
| $A_t$ | Adjacency matrix at time $t$. | $D_t$ | Degree matrix at time $t$. |
| $G_t$ | Graph snapshot at time $t$. | $\mathbf{X}_t$ | Node feature matrix at time $t$. |
| $\mathbf{H}_t^{\text{att}}$ | Temporal attention-based representation. | $\mathbf{H}_t$ | Final node representation at time $t$. |
| $\mathbf{H}_t^{\text{sp}}$ | Spatial node representation. | $f_\theta(\cdot)$ | Anomaly scoring network. |
| $\hat{y}_i^{(t)}$ | Predicted anomaly score for node $v_i$ at $t$. | $V^L$ | Labeled positive (anomalous) nodes. |
| $\sigma(\cdot)$ | Sigmoid activation function. | $V^U$ | Unlabeled nodes. |
| $R_P$ | Risk over labeled positives. | $R_U$ | Risk over unlabeled nodes. |
| $\alpha_U$ | EMA momentum parameter. | $\mathbb{E}[\cdot]$ | Expectation operator. |
| $R(f)$ | Risk for label distribution alignment. | $\hat{\mu}_t$ | Smoothed expectation of unlabeled nodes. |
| $\pi_P$ | Class prior probability of anomalies. | $\mathcal{L}_{\text{total}}$ | Overall training loss based on PU risk |

---

**Algorithm 1** WAVEN-PULL Training Workflow

---

**Require:** Dynamic graph $\mathcal{G} = \{G^{(1)}, G^{(2)}, \ldots, G^{(T)}\}$; class prior $\pi_P$; EMA momentum $\alpha_U$; labeled positives $V_L$; unlabeled nodes $V_U$
**Ensure:** Trained anomaly scoring network $f_\theta$
 1: **for** each snapshot $G^{(t)}$ for $t = 1$ to $T$ **do**
 2:    $\mathbf{H}_t^{\text{sp}} \leftarrow \text{BETAWAVELETGNN}(G^{(t)})$
 3:    $\mathbf{H}_t^{\text{att}} \leftarrow \text{TEMPORALATTENTION}(G^{(t)})$
 4:    Fuse representations: $\mathbf{H}_t \leftarrow \mathbf{H}_t^{\text{sp}} + \mathbf{H}_t^{\text{att}}$
 5: **end for**
 6: Build training batches $\mathcal{B}$ from $\{\mathbf{H}_1, \ldots, \mathbf{H}_{T-1}\}$ using $V_L$ and sampled $V_U$
 7: **for** each training epoch **do**
 8:    **for** each batch $(\mathbf{X}_P, \mathbf{X}_U) \in \mathcal{B}$ **do**
 9:       Predict scores: $\hat{y}_P \leftarrow f_\theta(\mathbf{X}_P), \quad \hat{y}_U \leftarrow f_\theta(\mathbf{X}_U)$
10:       Compute labeled set risk:   $R_P \leftarrow 1 - \mathbb{E}[\hat{y}_P]$
11:       Estimate and smooth unlabeled mean: $\hat{\mu}_t \leftarrow \alpha_U \cdot \hat{\mu}_{t-1} + (1 - \alpha_U) \cdot \mathbb{E}[\hat{y}_U]$
12:       Compute unlabeled risk: $R_U \leftarrow |\hat{\mu}_t - \pi_P|$
13:       Total loss:   $\mathcal{L}_t \leftarrow 2\pi_P \cdot R_P + R_U$
14:       Update model parameters: $\theta \leftarrow \theta - \eta \cdot \nabla_\theta \mathcal{L}_t$
15:    **end for**
16: **end for**
17: **return** Trained scoring network $f_\theta$

---

