# OpenReview forum: "Waven-Pull: Wavelet-based Anomaly Detection in Dynamic Graphs via Positive-Unlabeled Learning"
_ICLR.cc/2026/Conference — ICLR 2026 Conference Withdrawn Submission_

### Official Review · Reviewer_L6GF · 2025-10-27

**Soundness:** 2
**Presentation:** 1
**Contribution:** 2
**Rating:** 2
**Confidence:** 4

**Summary:**

This paper addresses the problem of anomaly detection in dynamic graphs, focusing on the challenging and practical Positive-Unlabeled (PU) learning setting where only a small number of positive (anomalous) labels are available. The authors identify two primary challenges: (1) the inherent over-smoothing of Graph Neural Networks (GNNs), which act as low-pass filters and suppress high-frequency anomaly signals , and (2) the prediction bias in PU settings, where models tend to classify all unlabeled nodes as normal.

To address these issues, the paper proposes WAVEN-PULL, a framework consisting of three main components. First, a dynamic graph encoder uses Beta-Wavelet Graph Convolution to capture multi-scale spectral patterns (including high-frequency signals) and temporal attention to model the evolution of node behaviors. Second, a PU-aware alignment module corrects prediction bias by aligning the predicted anomaly ratio on unlabeled data with a known class prior, stabilized by an Exponential Moving Average (EMA). Third, an anomaly probability estimation module maps the learned embeddings to anomaly scores. The authors provide extensive theoretical analyses in the appendix to justify their claims regarding oversmoothing, PU bias, and the stability of their method. Experiments on four real-world datasets show that WAVEN-PULL outperforms several baseline methods, achieving significant gains in AUC and ranking-based metrics。

**Strengths:**

1. The paper tackles a problem of high practical importance: anomaly detection in dynamic graphs under the realistic constraint of extreme label scarcity (the PU setting).

2. The paper is theoretically dense, providing a formal analysis of the "coupled amplification" of GNN over-smoothing and PU supervision bias (Appendix A.3.3) . It also provides detailed theoretical guarantees for the proposed encoder's ability to preserve high-frequency signals (Appendix B.1) and the PU-alignment module's optimality and stability (Appendix B.2) .

3. The experimental results are strong, showing substantial improvements over the selected baselines. The gains on ranking-based metrics like Precision@50 and Recall@50 are particularly noteworthy (Table 2)

**Weaknesses:**

1. Limited Methodological Novelty: The paper's core methodology appears to be a novel combination of existing components rather than a fundamental methodological advance. The use of wavelet-based GNNs to counter the low-pass filtering effect for anomaly detection is a well-established concept in static graphs (as cited by the authors ). The primary architectural contribution is the application of this static encoder to graph snapshots, followed by a standard node-level temporal attention mechanism. While the appendix provides justification for why this combination is theoretically sound, the architectural design itself remains incremental.

2. Weak Conceptual Integration: The paper does not convincingly articulate a deep, synergistic link between its two main technical components. The theoretical analysis argues that the temporal attention module preserves the spectral contrast captured by the wavelet encoder (Theorem B.5), but it fails to establish why a wavelet-based encoder is uniquely suited for or enhanced by the PU-aware learning objective. The framework currently reads as two effective but separate solutions addressing two separate problems (oversmoothing and PU bias), rather than a deeply integrated system where the components mutually amplify one another's benefits.

3. Insufficient Baseline Comparisons: The experimental evaluation is not comprehensive enough to establish state-of-the-art performance for a top-tier conference. The paper also lacks comparisons against several recent and powerful dynamic graph GNN architectures (e.g., PINT, DyGFormer, SILD) that are essential for contextualizing the performance of a new dynamic graph model. The ablation study for the PU-aware loss (-LDA in Figure 3)  only compares against a naive binary cross-entropy baseline, which is known to perform poorly in PU settings. A more rigorous evaluation would compare the proposed alignment loss against other established PU-learning losses (e.g., nnPU) while holding the wavelet-based graph encoder fixed.


4. The paper states that experiments were repeated five times , but the main tables (Table 1 and 2) report only average performance. Reporting standard deviations is critical for assessing the stability of the method and the statistical significance of its improvements.

5. The evaluation is conducted on datasets with a maximum of ~11,000 nodes. The generalizability and, more importantly, the scalability of the proposed method—which relies on spectral wavelet filters derived from the graph Laplacian —on larger, more realistic dynamic graphs (e.g., with millions of nodes) remain unevaluated.

**Questions:**

Can the authors elaborate on the synergistic connection between the Beta-Wavelet encoder and the PU-aware alignment module? Beyond temporal attention "not harming" the spectral properties (as shown in Appendix B.1.4 ), is there a reason why the PU-aware loss is particularly well-suited to optimizing the high-frequency representations produced by the wavelet encoder?

---

### Official Review · Reviewer_aQxo · 2025-10-29

**Soundness:** 3
**Presentation:** 3
**Contribution:** 2
**Rating:** 4
**Confidence:** 2

**Summary:**

This paper addresses the critical and highly practical problem of dynamic graph anomaly detection (DGAD) under the Positive-Unlabeled (PU) learning setting, where only a small subset of anomalies are explicitly labeled. The authors identify that conventional Graph Neural Networks (GNNs) fail in this setting due to a "coupled amplification" of GNN over-smoothing (which suppresses high-frequency anomaly signals) and PU supervision bias (which drives predictions toward the majority normal class, leading to model collapse). To counteract this, the paper proposes WAVEN-PULL, a unified framework featuring two core components: 1) a dynamic graph encoder that uses Beta-Wavelet Graph Convolution and temporal attention to capture multi-scale spectral patterns and preserve high-frequency information, and 2) a PU-aware alignment module that corrects prediction bias by enforcing consistency between the unlabeled prediction mean and the known anomaly class prior, stabilized via Exponential Moving Average (EMA). Extensive experiments on four real-world dynamic graph benchmarks demonstrate that WAVEN-PULL significantly outperforms state-of-the-art methods.

**Strengths:**

The paper presents a solid and well-motivated approach that tackles a truly challenging and practical scenario.

Originality & Significance (PU Setting): The work's primary strength lies in its explicit focus on the Positive-Unlabeled (PU) setting for dynamic graph anomaly detection. This scenario accurately reflects real-world constraints where large-scale, confirmed anomaly labels are extremely scarce. The formulation of the problem and the proposed solution's theoretical grounding make a significant contribution to practical DGAD.

Quality & Theoretical Rigor: The theoretical analysis is impressive. The paper provides a formal demonstration of the "coupled amplification" failure mode (Appendix A.3.3) inherent in standard GNNs under PU learning. Furthermore, the two main components are rigorously justified: Proposition B.2 proves that the Beta-Wavelet filter actively reduces the over-smoothing index, and Proposition B.9/Theorem B.11 establish the optimality and stability of the PU-aware alignment objective.

Clarity & Technical Solution: The proposed solution is a clean, dual fix to a dual problem. The encoder addresses the representational issue (suppressed high-frequency signals), while the PU-aware module addresses the supervision issue (loss bias and model collapse). The overall architecture is logical and effective, as supported by the strong experimental results across diverse datasets.

**Weaknesses:**

Limited Real-World Generality in PU Simulation: The experimental setting simulates the PU scenario by randomly sampling a fixed number (e.g., 100) of anomalies from the training set as labeled positives. While this is a common practice, it does not fully replicate the constraints of large-scale, real-world data where the labeled set is often biased. Random sampling provides an ideal scenario. It is questionable whether the method's effectiveness holds up when faced with a massive, highly contaminated unlabeled set where the available 100 labels are non-randomly sampled or represent only a tiny fraction of the detectable anomalies.

**Questions:**

The authors should discuss the limitations of their random sampling strategy and consider an experiment in the Appendix where the labeled positive set is chosen using a biased or non-random selection strategy (e.g., sampling anomalies with high degree centrality, low feature variance, or those that appeared only in earlier snapshots) to better mimic the discovery process in real systems. This would significantly strengthen the claim of robustness in the practical PU setting.

Since I am not an expert in this field, I would like to consider the opinions of the other reviewers before making my final decision.

---

### Official Review · Reviewer_Bu7c · 2025-10-31

**Soundness:** 4
**Presentation:** 3
**Contribution:** 3
**Rating:** 6
**Confidence:** 4

**Summary:**

This paper proposes WAVEN-PULL, a framework for anomaly detection in dynamic graphs under the Positive-Unlabeled (PU) learning setting. The work proposes two challenges: GNN over smoothing and PU supervision bias, then propose a two-pronged solution. The method consists of a dynamic graph encoder combining Beta-Wavelet Graph Convolution with temporal attention to capture multi-scale, high-frequency temporal patterns, and a PU-aware alignment module that provides an unbiased risk estimator by aligning predicted anomaly rates with a class prior.

**Strengths:**

S1. This work introduces a PU-aware alignment module that corrects prediction bias . This module is supported by extensive theoretical analysis showing it provides an unbiased risk estimator and ensures stability.

S2. The model uses Beta-Wavelet Graph Convolution combined with temporal attention to capture multi-scale spectral patterns. This specifically preserves the high-frequency signals indicative of anomalies.

S3. The paper shows consistent state-of-the-art results on several real-world dynamic graph benchmarks.

**Weaknesses:**

W1. The paper presents the methodology as a combination of several components, like a Beta-Wavelet GNN (existing work), a temporal attention mechanism (a standard technique), and a PU-aware risk estimator. The primary novelty appears to be the specific integration of these parts and the extensive theoretical justification that this combination is stable and optimal for this task. However, This presentation makes the novelty in the paper appear moderate.

W2. The paper states that the code "will be made available upon reasonable request." It is recommended authors be required to make their source code and experiment configurations publicly available in an anonymous repository for review and commit to a public release upon acceptance.

W3. It is recommended authors add a paragraph to the conclusion discussing the method’s limitations and proposing specific directions for future research.

**Questions:**

Q1. How were the baseline methods trained under the PU setting? Methods like TGAT and TGN are not familiar for anomaly detection or PU learning. If they were trained naively (treating all unlabeled nodes as negative), does this would be an unfair comparison?

---

### Official Review · Reviewer_g2Uq · 2025-11-01

**Soundness:** 2
**Presentation:** 2
**Contribution:** 2
**Rating:** 2
**Confidence:** 2

**Summary:**

This paper introduces WAVEN-PULL, a novel framework for identifying anomalous nodes in dynamic graphs under a Positive-Unlabeled (PU) learning regime. The authors motivate their work by highlighting two fundamental weaknesses of conventional GNNs in this context: their tendency to smooth over high-frequency signals indicative of anomalies due to their low-pass filtering nature, and their susceptibility to a prediction bias that favors the majority (normal) class when negative labels are absent. The proposed WAVEN-PULL architecture comprises three stages: (1) a spatiotemporal encoder that utilizes Beta-Wavelet Graph Convolution to preserve spectral information across multiple frequency bands and employs temporal attention to track node evolution; (2) a PU-aware alignment module designed to counteract supervision bias by regularizing the model's output distribution on unlabeled data to match a given class prior; and (3) a final prediction head to generate anomaly probabilities. The authors furnish a detailed theoretical appendix to support the design choices. Empirically, the method is shown to achieve compelling performance gains over existing techniques on four public benchmarks.

**Strengths:**

1.  The work addresses a relevant and challenging real-world task. The formulation of dynamic graph anomaly detection as a PU learning problem accurately reflects the operational constraints of many practical applications where confirmed anomaly labels are a scarce resource.
2.  The manuscript is supported by substantial theoretical derivations in the appendices. The authors provide a formal treatment of the interplay between GNN signal propagation and PU learning bias, as well as proofs concerning the spectral-preservation properties of the encoder and the optimality of the proposed learning objective.
3.  The empirical performance is compelling. The model demonstrates significant advantages over the selected baselines, particularly in its ability to correctly rank anomalous nodes, as evidenced by the strong Precision@50 and Recall@50 results reported in Table 2.

**Weaknesses:**

1.  The proposed method, while effective, appears to be an assemblage of pre-existing techniques rather than a foundational innovation. The core architectural components—Beta-Wavelet Graph Convolution for countering oversmoothing in anomaly detection and node-level temporal attention—are directly adapted from prior work. The contribution thus lies in the novel application and integration of these known elements to the dynamic PU setting, which constitutes an evolutionary rather than a revolutionary step.
2.  The paper presents two distinct solutions for two distinct problems (spectral signal loss and PU learning bias) but fails to weave them into a unified theoretical story. The justification provided is one of non-interference (i.e., the temporal module does not degrade the spatial module's spectral features) rather than true synergy (e.g., proving the PU objective is uniquely suited for wavelet-derived representations). This leaves the impression of a modular system where components are combined effectively but without a deep, unifying principle.
3.  The experimental validation is not sufficiently broad to claim state-of-the-art status.
    * The set of comparators is missing several high-impact, recent models in dynamic graph learning that would be necessary for a comprehensive benchmark.
    * The analysis of the PU learning module is not fully isolated. Its superiority is only demonstrated in the ablation study against a weak baseline (treating all unlabeled data as negative), rather than being rigorously compared to other principled PU learning objectives (e.g., nnPU) within the same architectural backbone.
4.  The empirical results lack statistical robustness. While the authors state that experiments were repeated five times, the results are presented as point estimates without standard deviations, making it impossible to assess the stability of the model or the statistical significance of the reported gains.
5.   The evaluation is confined to relatively small-scale graphs (the largest having ~11,000 nodes). The computational viability of the proposed spectral filtering approach, which involves polynomial operations on the graph Laplacian, on graphs with millions or billions of edges is not addressed, casting doubt on its applicability to large, industrial-scale problems.

**Questions:**

1.  Could the authors provide a more compelling narrative for the interplay between the wavelet-based encoder and the PU-aware objective? Is there a theoretical or empirical reason to believe that the representations from the multi-band spectral filters are especially amenable to optimization via the proposed alignment loss, beyond the general benefit of preserving anomaly signals?
2.  Could you provide a justification for not including more recent and powerful dynamic graph architectures in your baseline comparison to more rigorously contextualize the performance of WAVEN-PULL?

---

### Note · Authors · 2025-11-23

I have read and agree with the venue's withdrawal policy on behalf of myself and my co-authors.